# Beyond Expectations:
# Learning with Stochastic Dominance Made Practical

**Shicong Cen**                                                         *shicongc@andrew.cmu.edu*
*Carnegie Mellon University*

**Jincheng Mei**                                                                *jcmei@google.com*
*Google Research*

**Hanjun Dai**                                                                  *hadai@google.com*
*Google Research*

**Dale Schuurmans**                                                       *schuurmans@google.com*
*Google Research & University of Alberta*

**Yuejie Chi**                                                               *yuejie.chi@yale.edu*
*Yale University*

**Bo Dai**                                                                      *bodai@google.com*
*Google Research & Georgia Tech*

**Reviewed on OpenReview:** *https://openreview.net/forum?id=ebyPKXsweD*

## Abstract

Stochastic dominance serves as a general framework for modeling a broad spectrum of decision preferences under uncertainty, with risk aversion as one notable example, as it naturally captures the intrinsic structure of the underlying uncertainty, in contrast to simply resorting to the expectations. Despite theoretical appeal, the application of stochastic dominance in machine learning has been scarce, due to the following challenges: **i)**, the original concept of stochastic dominance only provides a *partial order*, and therefore, is not amenable to serve as a general optimality criterion; and **ii)**, an efficient computational recipe remains lacking due to the continuum nature of evaluating stochastic dominance.

In this work, we make the first attempt towards establishing a general framework of learning with stochastic dominance. We first generalize the stochastic dominance concept to enable feasible comparisons between any arbitrary pair of random variables. We next develop a simple and computationally efficient approach for finding the optimal solution in terms of stochastic dominance, which can be seamlessly plugged into many learning tasks. Numerical experiments demonstrate that the proposed method achieves comparable performance as standard risk-neutral strategies and obtains better trade-offs against risk across a variety of applications including supervised learning, reinforcement learning, and portfolio optimization.

# 1 Introduction

In machine learning and operations research, the prevalent paradigm of decision making in the presence of uncertain and stochastic outcomes is to maximize (resp. minimize) the expected utility (resp. loss) with respect to the decision variables. However, the expectation of the decision-dependent utility function *alone* often depicts an overly simplified snapshot of its *distribution*, ignoring the intrinsic structure of the underlying uncertainty. As such, it fails to distinguish decisions with the same expected utilities but drastically different outcomes or model behaviors, especially when taking *risk* into consideration.

There are no shortage of *risk-sensitive* applications where taming the risk is at least as important as maximizing the utility, examples including financial planning, medical examinations, robotics and autonomous systems, to mention a few. In these high-stake applications, the principle of expectation may lead to inferior decisions due to its uncertainty-agnostic nature. To motivate our discussions, we showcase three distinct applications where risk-averse solutions are of particular interest, which will run throughout this paper.

- *Risk-sensitive supervised learning.* In standard supervised learning, one aims to find an optimally parameterized model such that the expected loss, which measures the difference between the model output and the target output given an input feature, is minimized. However, excessive prediction errors, even with extreme low probability, can pose significant risk to the system operation which may be undesirable.

- *Risk-sensitive reinforcement learning.* Reinforcement learning (RL) formulates sequential decision making problems as Markov decision processes (MDPs). The goal is to design an action selection rule, i.e., a policy, which maximizes the expected cumulative reward collected over the trajectories by executing the policy. Nonetheless, risky actions, if not discouraged properly, might still be deployed when they are compensated by high reward in the long horizon.

- *Portfolio optimization.* A leading example in finance planning is the selection between mutually exclusive investment opportunities or portfolios with uncertain returns. The allocation of the assets yields a random variable representing the total return. Maximizing the expected total return alone can lead to large volatility and increases the probability of suffering significant loss.

## 1.1 Learning with Stochastic Dominance

One needs to go *beyond expectations* to handle risk, a topic that has been extensively researched in many disciplines. Broadly, existing approaches can be categorized into two paradigms. The first relies on a scalar metric that induces a total order over outcomes, thereby allowing every pair of random variables to be compared. The mean–risk (Markowitz & Todd, 2000) framework is a notable example: it quantifies the problem with two metrics: a *mean* that measures the expected outcome, and a *risk* that measures the variability of outcomes. Popular choices of risk measures include variance (Markowitz & Todd, 2000), semideviation (Ogryczak & Ruszczyński, 1999), entropic risk (Rudloff et al., 2008), and so on. The mean-risk approach models risk-averse preferences by penalizing the mean with the risk measure and allows simple trade-offs and efficient learning algorithms (Maurer & Pontil, 2009; Duchi & Namkoong, 2019). However, the design choices of the risk measure and corresponding trade-offs are usually ad-hoc, lacking rigorous justifications. In contrast, the second paradigm leverages partial orders to capture more fine-grained preference structures, among which stochastic dominance (Mann & Whitney, 1947) is the most prominent.

**Stochastic dominance.** Stochastic dominance (SD) (Mann & Whitney, 1947; Lehmann, 1955) provides a principled scheme of comparing real-valued random variables by considering the *full spectrum* of their $k$th-order cumulative distribution functions, instead of condensing into a single scalar metric. In fact, efficient solutions found by the mean-risk approach can be stochastically dominated by other feasible solutions (Ogryczak & Ruszczyński, 2002), suggesting SD offers stronger guidance and finer granularity in modeling the risk-averse preference. In addition, the deployment of SD for comparing random variables does not need additional assumptions on the distribution (e.g., the mean-variance approach requires normality (Levy, 2015)).

Another nice justification of SD comes from expected utility theory (Boutilier et al., 2006; Armbruster & Delage, 2015). Specifically, if one solution stochastically dominates the other, it yields higher expected utility for *any* utility in a wide class of functions (e.g., non-decreasing functions for first-order dominance). Since SD implies higher expectation *in the risk-neutral sense* by setting the utility function as the identity function, SD is more selective in model selection as a risk-averse criterion, without the need to specify utility functions.

**Challenges.** Despite the appealing theoretical properties of SD, applications of SD in machine learning remain scarce. In truth, practical algorithms for finding a desirable solution under the criterion of SD remain lacking due to the following critical challenges:

- SD, in its existing form, only defines a *partial order* over all distributions of real-valued random variables, which is unsuitable to be optimized directly.

- Evaluating SD involves comparisons along a *continuum* of cumulative distribution functions and thus necessitates computationally efficient algorithms.

Existing literature on the computational aspects of stochastic dominance circumvents the first challenge by studying stochastic dominance constraint optimization (Dai et al., 2023), with the goal of maximizing the objective function — typically simply set to the expectation of the utility — over a feasible set that consists of all dominating solutions versus a predefined reference solution. However, the fact that the reference solution being fixed compromises the optimality guarantee in the first place, as the approach fails to distinguish two feasible solutions with the same expected utility, even if one dominates the other. On the other end, while it is possible for standard optimization formulations over a global scalar metric to yield an optimal SD solution (e.g., optimizing a nondecreasing and strictly concave utility function guarantees non-dominance in the second order), the induced total order introduces extra preference beyond SD that can be undesirable. This motivates the question:

*Can we design a practical algorithm that finds an optimal solution in terms of stochastic dominance?*

## 1.2 Our contribution

In this work, we aim to establish a general framework of learning with stochastic dominance, by tackling the two challenges mentioned above. We handle the first challenge by quantifying the degree of stochastic dominance as a functional and formulating SD optimality as a fixed point of the corresponding optimization process. This motivates the design of an iterative optimization procedure with non-stationary objective functions that can be solved efficiently. We summarize our contributions as follows.

- We first generalize the original stochastic dominance concept to enable feasible comparisons between any arbitrary pair of random variables, paving the way to a general machine learning framework that optimizes stochastic dominance.

- We propose **L**earning with **S**tochastic **D**ominance (LSD), a novel first-order method for finding approximate optimal solutions in terms of stochastic dominance in the hypothesis space. The optimality under LSD is equivalent to non-dominance under SD and hence allows direct combination with any risk measure or preference functional that is compatible with SD.

- We establish convergence guarantees under mild technical assumptions despite the non-stationary nature of the optimization process. It is shown that LSD finds an $\epsilon$-approximate optimal solution within $\mathcal{O}(\epsilon^{-2})$ iterations, which introduces minimal computational overhead compared with standard mini-batch stochastic gradient method.

- We draw connections between SD and distributionally robust optimization (DRO), allowing us to interpret the proposed method as optimizing a surrogate distributionally robust loss.

To the best of our knowledge, this work presents the first attempt towards a computationally tractable approach for learning stochastic dominance optimal solutions, both practically and theoretically. Numerical experiments are demonstrated to illustrate the effectiveness of our framework for finding risk-averse yet performant solutions in a variety of learning tasks such as supervised learning, reinforcement learning, and portfolio optimization.

The rest of this paper is organized as follows. Section 2 develops a general learning framework using stochastic dominance. Section 3 presents a computationally efficient algorithm and its theoretical computational complexity. Numerical results are presented in Section 4. Finally, we conclude the paper in Section 5, and leave the proofs and a connection to DRO in Appendix A.

**Notation.** We denote real-valued random variables by upper case letters, e.g., $X$, and the corresponding observed values by lower case letters, e.g., $x$. The set of probability measures over set $\mathcal{A}$ is denoted by $\Delta(\mathcal{A})$. $(x)_+$ is a shorthand notation of $\max(0, x)$. Two random variables $X$ and $Y$ are equal in distribution if they have the same distribution, denoted as $X \overset{\mathsf{D}}{=} Y$.

### 1.3 Related works

**Stochastic-dominance constrained optimization.** In the literature, SD is often used to characterize the feasible set of an optimization problem as a constraint w.r.t. a *given* competitor. In contrast, in our SD learning framework, we seek the optimal solution in the stochastic dominance sense within the *whole hypothesis space*, instead of against a fixed competitor. Many previous related works tackle the SD constrained optimization by casting the comparison of $k$th-order cumulative distribution functions (Dentcheva & Ruszczynski, 2003; Dentcheva & Ruszczyński, 2004; Noyan et al., 2006) or its equivalent reformulations (Luedtke, 2008; Post, 2003; Armbruster & Delage, 2015) as linear programming and mixed-integer programming problems, which typically incurs a quadratic iteration complexity/memory consumption and is hence not applicable to large-scale practical problems. Dai et al. (2023) presents the latest effort towards efficiently solving SD constrained optimization and achieves (near) linear computation and memory cost. The key ingredient lies in the efficient solver for the inner optimization in the Lagrangian, which is integral to our learning algorithm development as well.

**Connections to other risk-sensitive approaches.** Quantile statistics such as Value at Risk ($\text{VaR}_\alpha$) (Markowitz, 1952; Roy, 1952) and Conditional Value at Risk ($\text{CVaR}_\alpha$) (Artzner et al., 1999) represent another popular choice of risk measure beyond variance. For a risk level $\alpha \in (0, 1)$, $\text{VaR}_\alpha$ is given by the $(1 - \alpha)$-quantile of the loss, whereas $\text{CVaR}_\alpha$ takes a step further by focusing on the conditional expectation of loss beyond $\text{VaR}_\alpha$. Remarkably, second-order stochastic dominance (SSD) can be interpreted as a continuum of $\text{CVaR}_\alpha$ comparison over the entire risk level set $(0, 1)$ (Martin et al., 2020), which again justifies the superior theoretical properties of SD.

Beyond the uncertainty that stems from a known data distribution, distributionally robust optimization (DRO) seeks to optimize the model against the uncertainty in the knowledge of the distribution itself, by focusing on the worst-case expectation under some distribution shift. Duchi et al. (2021) demonstrated that for uncertainty set induced from $f$-divergence balls, the DRO formulation is asymptotically equivalent to a mean-risk treatment, with the risk measure given by the square root of variance.

**Other generalization of stochastic dominance.** Notable prior efforts of generalizing stochastic dominance and mitigating incomparability include *almost stochastic dominance* (Leshno & Levy, 2002), *statistical preference* (De Schuymer et al., 2003), *graded stochastic dominance* (Pérez-Fernández et al., 2021), *generalized stochastic dominance for classifier comparison* (Jansen et al., 2023) with further developments in Müller et al. (2025); Luo et al. (2017); De Schuymer et al. (2005); Jansen et al. (2024), among others. However, these contributions remain largely theoretical and do not yield a practical machine learning framework that is compatible with stochastic dominance, which we seek to address in this work.

**Distributional RL.** Distributional RL (Bellemare et al., 2023) provides a systematic approach towards learning the distribution of the cumulative rewards induced by executing a policy in RL. While this allows

the decision maker to resort to mean-risk approaches for risk-averse policies, a policy improvement scheme that fully utilizes the learned distributions in terms of stochastic dominance remains lacking. Martin et al. (2020) investigated the use of SD for action selection using a particle-based algorithm, which involved extra computation and thus fell short of providing an explicit policy.

## 2 Stochastic Dominance Learning

In this section, we first introduce the concept of stochastic dominance, and reveal the difficulty in defining optimality in terms of stochastic dominance. We then resolve this difficulty and establish the stochastic dominance learning framework.

**Stochastic dominance.** Let $X$ denote a real-valued random variable. The $k$-th distribution function $F_X^k$ is defined recursively as

$$F_X^1(\eta) = \mathbb{P}_X(X \leqslant \eta); \tag{1}$$

$$F_X^k(\eta) = \int_{-\infty}^{\eta} F_X^{k-1}(\alpha)\, d\alpha = \frac{1}{(k-1)!}\mathbb{E}_X\left[(\eta - X)_+^{k-1}\right], \qquad k \geqslant 2 \tag{2}$$

where $F_X^1(\eta)$ is simply the standard cumulative distribution function (CDF). Then, $X$ dominates $Y$ in the $k$-th-order if (Mann & Whitney, 1947; Dentcheva & Ruszczynski, 2003; Lehmann, 1955)

$$F_X^k(\eta) \leqslant F_Y^k(\eta), \quad \forall \eta \in \mathbb{R}, \tag{3}$$

denoted as $X \succeq_k Y$. By definition, the $k$-th-order dominance implies the $(k+1)$th-order dominance. In practice, the popular choices are $k = 1$ or $k = 2$. First-order stochastic dominance (FSD), by definition, pursues consistently a lower probability of the random variable falling below a threshold, which equivalently asserts the existence of $\overline{X} \stackrel{D}{=} X$ and $\overline{Y} \stackrel{D}{=} Y$ such that $\overline{X} \geqslant \overline{Y}$ a.s. Second-order stochastic dominance (SSD), due to the aforementioned intrinsic relationship to CVaR, allows more fine-grained comparisons among random outcomes with the same expectations. Figure 1 illustrates two normal distributions centered at 0 with different variance, and their corresponding $F^2$ function. SSD favours the one with smaller variance as it yields a consistently lower $F^2$ function.

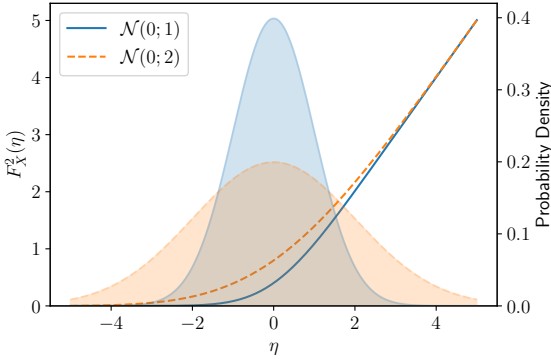

Figure 1: Probability density and second-order CDF of $\mathcal{N}(0;1)$ and $\mathcal{N}(0;2)$.

**Learning with SD.** In many applications, we can model the quantity of interest by $X_\theta$, associated with some parameterized model $\theta \in \Theta$, which we would like to maximize in general. As examples, we can rethink a few typical machine learning and decision making problems through the lens of random variable selection, to which we hope to leverage SD to reason their preferences.

- *Supervised learning.* Let feature vector $\mathbf{x}$ and corresponding label y be sampled from data distribution $\mathcal{D}$. The performance of a parameterized model $f_\theta$ is measured by the sample loss function

$\ell(f_\theta(\mathbf{x}), y))$. Typical choices of $\ell$ include mean squared error for regression tasks and cross entropy loss for classification tasks. Empirical "risk" minimization seeks to optimize the expected sample loss $\mathbb{E}_{(\mathbf{x}, y) \sim \mathcal{D}} \ell(f_\theta(\mathbf{x}), y))$ by minimizing the empirical estimate $\frac{1}{N} \sum_{i=1}^{N} \ell(f_\theta(\mathbf{x}_i), y_i)$, where $\{\mathbf{x}_i, y_i\}_{i=1}^{N}$ is the training dataset. We negate the sample loss in consistency with the definition of stochastic dominance, i.e., $X_\theta = -\ell(f_\theta(\mathbf{x}), y), (\mathbf{x}, y) \sim \mathcal{D}$. One can compare different models via examining SD relations among $X_\theta$'s over the hypothesis space of $\theta$.

- *Reinforcement learning.* Consider a discounted MDP with state space $\mathcal{S}$, action space $\mathcal{A}$, reward function $r : \mathcal{S} \times \mathcal{A} \to \mathbb{R}$ and transition kernel $P(\cdot|s, a)$ that defines the distribution of the next state upon choosing action $a$ at state $s$. A policy $\mathcal{S} \mapsto \Delta(\mathcal{A})$ defines a random action selection rule for each observed state, which induces random trajectory $\tau = (s_0, a_0, r_0, s_1, \cdots)$ where $a_t \sim \pi(\cdot|s_t)$, $s_{t+1} \sim P(\cdot|s_t, a_t)$, $r_t = r(s_t, a_t)$ from some initial state $s_0 \in \mathcal{S}$. The cumulative reward of a trajectory $\tau$ is given by $r(\tau) = \sum_{t=0}^{\infty} \gamma^t r_t$, where $\gamma \in (0, 1)$ is the discount factor. Standard learning paradigm in RL is to maximize the expected cumulative return $\mathbb{E}_\tau[r(\tau)]$, i.e., value function, with respect to a parameterized policy $\pi_\theta$. Stochastic dominance for comparison between policies can be therefore deployed by setting $X_\theta = r(\tau), \tau \sim P \circ \pi_\theta$.

- *Portfolio optimization.* Let random variable $R_i$ denote the return of stock $i$ which can be heavy-tailed and correlated. Portfolio optimization seeks to get an ideal allocation $w_\theta \in \Delta_K$ of assets among $K$ stocks to achieve trade-off between the expected return $\mathbb{E}[\sum_{i=1}^{K} [w_\theta]_i R_i]$ and some risk measure. Stochastic dominance naturally applies by focusing on the total return $X_\theta = \sum_{i=1}^{K} [w_\theta]_i R_i$.

**Generalized stochastic dominance for optimality.** One might be tempted to search for a model $\theta^\star$ that dominates all $\theta \in \Theta$, i.e., the *greatest element* under stochastic dominance rule in the aforementioned learning scenarios, which would imply (c.f. (3))

$$\min_\eta \left[ F_{X_\theta}^k(\eta) - F_{X_{\theta^\star}}^k(\eta) \right] \geqslant 0, \quad \forall \theta \in \Theta. \tag{4}$$

However, such $\theta^\star$ is not guaranteed to exist due to the fact that SD only defines *partial* order among distributions. In other words, there exist two random variables such that their order cannot be distinguished in the sense of SD. Therefore, it is impossible to define a general *"optimality"* in the sense of (4). This gap hinders the development of a learning framework under SD from both theoretical justification and optimization-based algorithm design, which motivates a more general definition of SD. Consider the following two statements:

1. $X_{\theta^\star}$ is a *maximal element* under the partial order of stochastic dominance, i.e., not dominated by other feasible solutions.

2. We have

$$\max_\eta \left[ F_{X_\theta}^k(\eta) - F_{X_{\theta^\star}}^k(\eta) \right] \geqslant 0, \quad \forall \theta \in \Theta, \tag{5}$$

where the equality is achieved only when $X_\theta \stackrel{\mathsf{D}}{=} X_{\theta^\star}$.

In view of the equivalence of the above two statements, we propose the following *Generalized Stochastic Dominance* functional:

$$\Omega_k(X, Y) = \max_{\eta \in [a, b]} \left[ F_X^k(\eta) - F_Y^k(\eta) \right], \tag{6}$$

which quantifies the degree of stochastic dominance between $X$ and $Y$ over the interval $[a, b]$ in an unilateral way.[1] Here, we follow the convention of enforcing stochastic dominance on an interval $[a, b]$ to overcome numerical tractability issues and technical subtleties (Leshno & Levy, 2002; Dentcheva & Ruszczynski, 2003). The proposition states the existence of such non-dominated solutions under mild conditions, with the proof deferred in Appendix C.

---

[1] With slight abuse of notation, we drop the dependency on $[a, b]$ in the notation of $\Omega$ for conciseness.

**Proposition 1** *It is guaranteed that a non-dominated solution $\theta^\star$ exists as long as $\Theta$ is compact and that $F_{X_\theta}^k(\eta)$ is continuous with regard to $\theta$ for every $\eta \in \mathbb{R}$.*

In view of the generalized SD in (5), it is now natural to define a general learning problem through an optimization lens, by seeking an approximate optimal solution $\widehat{\theta}^\star$ such that for any $\theta \in \Theta$, it holds that

$$\Omega_k(X_\theta, X_{\widehat{\theta}^\star}) \geqslant -\epsilon. \tag{7}$$

In other words, $\widehat{\theta}^\star$ is guaranteed to not be dominated by any other solution $\theta$ by a margin of $\epsilon$ over the interval $[a, b]$.

## 3 LSD: First-order Optimization for Learning with SD

In this section, we design efficient first-order algorithm to solve (7), resolving the computational difficulty discussed in Section 1.

### 3.1 Stochastic Gradient for SD Learning

The optimality condition (5) can be written as

$$\theta^\star = \arg\min_\theta \Omega_k(X_\theta, X_{\theta^\star}),$$

which motivates us to interpret $\theta^\star$ as a fixed point of the following iterative process:

$$\theta_{t+1} \leftarrow \arg\min_\theta \Omega_k(X_\theta, X_{\theta_t}). \tag{8}$$

Here, $\theta_t$ denote the choice of parameter $\theta$ at the $t$-th iteration. Rather than pursuing an exact optimal solution to (8), we show that it suffices to find $\theta_{t+1}$ such that

$$\Omega_k(X_{\theta_{t+1}}, X_{\theta_t}) < -\epsilon, \tag{9}$$

since failing to find such $\theta_{t+1}$ implies the approximate optimality of $\theta_t$ by definition (7). Since $\Omega_k(X_{\theta_t}, X_{\theta_t}) = 0$, this amounts to making progress of $\epsilon$ on the sub-problem

$$\min_\theta \Omega_k(X_\theta, X_{\theta_t}), \tag{10}$$

which makes gradient-based methods an ideal candidate.

**Subgradient calculation.** It remains unclear the optimization properties of (10) as well as how to estimate gradients. To proceed, we shall resort to the utility reformulation of $\Omega$. Note that $\Omega_k(X, Y)$ can be equivalently written in a variational form:

$$\Omega_k(X, Y) = \max_{\mu \in \Delta([a,b])} \int_a^b \left[ F_X^k(\eta) - F_Y^k(\eta) \right] \mathrm{d}\mu(\eta), \tag{11}$$

where the maximum is taken over probability measures over $[a, b]$. For every choice of $\mu$, changing the order of integral by Fubini's theorem, we have

$$\int_a^b \left( F_X^k(\eta) - F_Y^k(\eta) \right) \mathrm{d}\mu(\eta)$$
$$= \int_a^b \frac{1}{(k-1)!} \left[ \int_{-\infty}^\infty (\eta - x)_+^{k-1} f_X(x) \mathrm{d}x - \int_{-\infty}^\infty (\eta - y)_+^{k-1} f_Y(y) \mathrm{d}y \right] \mathrm{d}\mu(\eta)$$
$$= \frac{1}{(k-1)!} \left[ \int_{-\infty}^\infty \int_a^b (\eta - x)_+^{k-1} f_X(x) \mathrm{d}\mu(\eta) \mathrm{d}x - \int_{-\infty}^\infty \int_a^b (\eta - y)^{k-1} f_Y(y) \mathrm{d}\mu(\eta) \mathrm{d}y \right]$$

$$= -\int_{-\infty}^{\infty} u(x) f_X(x) \mathrm{d}x + \int_{-\infty}^{\infty} u(y) f_Y(y) \mathrm{d}y$$

$$= \underbrace{-\mathbb{E}_X[u(X)] + \mathbb{E}_Y[u(Y)]}_{=:L(X,Y,u)}, \tag{12}$$

where the utility function $u$ is defined as

$$u(x) = -\frac{1}{(k-1)!} \int_a^b (\eta - x)_+^{k-1} \mathrm{d}\mu(\eta). \tag{13}$$

Therefore, we can write $\Omega_k(X, Y)$ as

$$\Omega_k(X, Y) = \max_{u \in \mathcal{U}_k} \Big\{ -\mathbb{E}_X[u(X)] + \mathbb{E}_Y[u(Y)] \Big\}.$$

Here, $\mathcal{U}_k = \{u : u(x) = -\frac{1}{(k-1)!} \int_a^b (\eta - x)_+^{k-1} \mathrm{d}\mu(\eta), \mu \in \Delta([a, b])\}$ collects all utility functions that can be expressed in the form of (13).

Note that when $k \geqslant 2$, $u \in \mathcal{U}_k$ is non-decreasing and concave, which guarantees $u(x_\theta)$ to be concave with regard to $\theta$, as long as $x_\theta$ is concave (Boyd & Vandenberghe, 2004). When the sampling probability of $X_\theta$ is independent of $\theta$, such as in supervised learning and portforlio optimization, $\Omega_k(X_\theta, Y)$ takes maximum over a set of convex functions and is therefore convex as well. The subgradients of $\Omega_k(X_\theta, Y)$ (Bertsekas, 1971) is given by

$$\partial_\theta[\Omega_k(X_\theta, Y)] = \mathrm{conv}\Big\{ -\partial_\theta\big[\mathbb{E}_{X_\theta}[u(X_\theta)]\big] : u \in \mathcal{U}_k^\star \Big\}$$

$$= \mathrm{conv}\Big\{ -\mathbb{E}_{X_\theta}[\partial_\theta u(X_\theta)] : u \in \mathcal{U}_k^\star \Big\},$$

where $\mathcal{U}_k^\star = \arg\max_{u \in \mathcal{U}_k} L(X, Y, u)$, and conv is the convex hull. The expectation formulation of the above equation allows estimation of the subgradient using sampling, i.e., the sample average, and subgradient chain rule (see e.g., Clason & Valkonen (2020, Theorem 4.19)), given by

$$-\frac{1}{N} \sum_{i=1}^N \partial_\theta u(x_i) = -\frac{1}{N} \sum_{i=1}^N \partial_{x_i} u(x_i) \partial_\theta x_i, \tag{14}$$

where $\{x_i\}_{i=1}^N$ are $N$ data points sampled from $X_\theta$. Here, we omit the dependency with $\theta$ in $x_i$ for notational simplicity. This allows interpreting our proposed method as stochastic gradient methods with each sample $x_i$ *dynamically weighted* by $\partial_{x_i} u(x_i)$. For learning tasks with model-dependent sampling probability (e.g., RL), one can instead apply log-derivative trick (Williams, 1992) for gradient estimation (see Appendix B for more details).

**Final algorithm.** We summarize the algorithm procedure in Algorithm 1. Simply put, the algorithm follows a nested-loop design, where the inner loop focuses on solving (10) by first obtaining $\hat{u}^\star$ that maximizes the sample estimate of $L$ and then derive the stochastic subgradient with (14). We terminate the inner loop and update $\theta_t$ when the progress condition (9) is approximately met. If (9) is not met within a certain number of iterations, we conclude that the current $\theta_t$ is approximately optimal and return the solution.

### 3.2 Theoretical Analysis

Two questions arise naturally with regard to the theoretical guarantee of the proposed method: **i)**, whether it is guaranteed to converge, and **ii)**, whether it induces a significantly higher iteration complexity compared with standard minibatch SGD methods. The concern stems from the fact that the dynamics of Algorithm 1 cannot be interpreted as an optimization process targeting a fixed objective function, and that one round of inner loop alone can take $\mathcal{O}(\epsilon^{-2})$ iterations to end.

The following theorem addresses the concerns mentioned above by ensuring convergence within $\tilde{\mathcal{O}}(\epsilon^{-2})$ total iteration complexity.

---

**Algorithm 1:** Learning with Stochastic dominance (LSD)

---

**1** **Input:** Initialization $\theta_0$.
**2** **for** $t = 0, \cdots, T_{max} - 1$ **do**
**3**      Set $\theta_{t,0} = \theta_t$.
**4**      **for** $\bar{t} = 0, \cdots, \overline{T}_{max} - 1$ **do**
**5**          Sample data $\{x_{t,\bar{t},i}\}_{i=1}^N \sim X_{\theta_{t,\bar{t}}}^N$ and $\{x_{t,i}\}_{i=1}^N \sim X_{\theta_t}^N$.
**6**          Compute $\widehat{u}^\star = \underset{u \in \mathcal{U}_k}{\operatorname{argmax}} \, \widehat{L}(X_{\theta_{t,\bar{t}}}, X_{\theta_t}, u)$, where

$$\widehat{L}(X_{\theta_{t,\bar{t}}}, X_{\theta_t}, u) := -\frac{1}{N}\sum_{i=1}^N u(x_{t,\bar{t},i}) + \frac{1}{N}\sum_{i=1}^N u(x_{t,i}).$$

         Update $\theta_{t,\bar{t}+1} = \theta_{t,\bar{t}} - \eta_{\bar{t}} g_{t,\bar{t}}$, where

$$g_{t,\bar{t}} \in -\frac{1}{N}\sum_{i=1}^N \partial_{x_{t,\bar{t},i}} \widehat{u}^\star(x_{t,\bar{t},i}) \partial_\theta x_{t,\bar{t},i}.$$

         **if** $\widehat{\Omega_k}(X_{\theta_{t,\bar{t}+1}}, X_{\theta_t}) \leqslant -\epsilon/2$ **then**
**7**              Set $\theta_{t+1} = \theta_{t,\bar{t}+1}$.
**8**              **Break**.
**9**      **if** $\theta_t$ *is not updated* **then**
**10**          **Return** $\theta_t$.

---

**Theorem 2** *For second-order stochastic dominance ($k = 2$), assume that $x_\theta$ is concave with regard to $\theta$, and bounded subgradient $\|g_{t,\bar{t}}\|_2^2 \leqslant G^2$ and bounded $k-$th order CDF $F_k(X_\theta, \eta) \leqslant C, \forall \eta \in [a, b]$. Let $\eta_{\bar{t}} = 1/\sqrt{\bar{t}}$, and sample size $N = \widetilde{\mathcal{O}}(\epsilon^{-2})$, $T_{max} = \lceil 4C/\epsilon + 1 \rceil$, $\overline{T}_{max} = \widetilde{\mathcal{O}}(\epsilon^{-2})$. For any initialization $\theta_0$, with probability $1 - \delta$, Algorithm 1 finds $\theta_t$ such that for any $\theta$,*

$$\Omega_k(X_\theta, X_{\theta_t}) \geqslant -\epsilon$$

*within $\widetilde{\mathcal{O}}(\epsilon^{-2})$ iterations.*

Several remarks are in order.

- The approximation error $\epsilon$ incorporates the statistical error due to sampling that scale with $N^{-1/2}$, which necessitates a choice of $N = \widetilde{\mathcal{O}}(\epsilon^{-2})$, similar to the case for empirical risk minimization.

- The iteration complexity of $\widetilde{\mathcal{O}}(\epsilon^{-2})$ is on par with that of subgradient methods for optimizing non-smooth convex functions, which sets LSD as an appealing alternative to standard risk-neutral approaches in practice for risk-averse applications. The iteration complexity can be further improved by incorporating regularization terms of $\mu$, e.g., entropy regularization, in the variational form (11) to ensure the uniqueness of the maximizer $\mu^\star$, which leads to differentiability of $\Omega$ by Danskin's theorem.

- While we state the theorem for $k = 2$ for simplicity, the analysis can be easily generalized to $k \geqslant 2$ by adopting the Rademacher complexity of $\mathcal{U}_k$ and upper bound of $F_k$ accordingly.

- Algorithm 1 can be used plug-and-play for achieving non-dominance in conjunction with standard empirical risk minimization, CVaR optimization, and related procedures, as long as the underlying risk or preference measure is consistent with stochastic dominance.

---

**Algorithm 2:** Utility solver for $k = 2$

---

**1** **Input:** samples $\{x_i\}_{i=1}^N$ and $\{y_i\}_{i=1}^N$.

**2** Sort both sequences of samples in increasing order, and merge them into $\{\eta_i\}_{i=1}^{2N}$.

**3** **for** $i = 1, \cdots, 2N$ **do**

**4**

$$\widehat{F}_X^1(\eta_i) = \widehat{F}_X^1(\eta_{i-1}) + \mathbf{1}_{\eta_i \in \{x_i\}}/N,$$
$$\widehat{F}_Y^1(\eta_i) = \widehat{F}_Y^1(\eta_{i-1}) + \mathbf{1}_{\eta_i \in \{y_i\}}/N,$$
$$\widehat{F}_X^2(\eta_i) = \widehat{F}_X^2(\eta_{i-1}) + (\eta_i - \eta_{i-1})\widehat{F}_X^1(\eta_{i-1}),$$
$$\widehat{F}_Y^2(\eta_i) = \widehat{F}_Y^2(\eta_{i-1}) + (\eta_i - \eta_{i-1})\widehat{F}_Y^1(\eta_{i-1}).$$

**5** Get $\widehat{\mu}^\star \in \Delta\Big( \arg\max_{\eta_i \in [a,b]} \big( \widehat{F}_X^2(\eta_i) - \widehat{F}_Y^2(\eta_i) \big) \Big)$.

**6** **for** $i = 2N, \cdots, 1$ **do**

**7**

$$\widehat{u}_1(\eta_i) = \widehat{u}_1(\eta_{i+1}) - \widehat{\mu}^\star(\eta_i),$$
$$\widehat{u}_2(\eta_i) = \widehat{u}_2(\eta_{i+1}) + (\eta_{i+1} - \eta_i)\widehat{u}_1(\eta_{i+1}).$$

**8** **Return** $\widehat{u}_2$.

---

A key ingredient of our analysis is to relate the sub-optimality gap $\min_\theta \Omega_k(\theta, \theta_t)$ in the $t$th loop with the optimization progress in the subsequent rounds, despite the fact that they are associated with different objective functions. This is made possible by exploiting a triangular inequality with $\Omega$, which ensures that the inner loop generally takes a smaller number of iterations than $\overline{T}_{\texttt{max}}$, ensuring the final iteration complexity is still $\widetilde{\mathcal{O}}(\epsilon^{-2})$. The proof is postponed to Appendix D.

### 3.3 Practical Implementation

When $k \leqslant 3$, the computation of $\widehat{u}^\star$ can be done in an efficient way that consumes $\mathcal{O}(N)$ memory and $\widetilde{\mathcal{O}}(N)$ time (Dai et al., 2023). Below we demonstrate the case with $k = 2$. Recall that each candidate utility function $u$ is associated with a probability measure $\mu$ by

$$u(x) = -\mathbb{E}_{\eta \sim \mu}[(\eta - x)_+]. \tag{15}$$

For $\widehat{L}(X, Y, u)$ induced by samples $\{x_i\}_{i=1}^N$ and $\{y_i\}_{i=1}^N$, we still have $\widehat{L}(X, Y, u) = \int_a^b \big( \widehat{F}_X^2(\eta) - \widehat{F}_Y^2(\eta) \big) \mathrm{d}\mu(\eta)$, where $\widehat{F}_X^2(\eta)$ is an empirical estimate of $F_X^2(\eta)$ given by $\widehat{F}_X^2(\eta) = \frac{1}{N} \sum_{i=1}^N (\eta - x_i)_+$. Note that $\widehat{F}_X^2(\eta) - \widehat{F}_Y^2(\eta)$ is piece-wise linear and hence achieves its maximum among the sample values $\{x_i\}$ and $\{y_i\}$. It is then straightforward to get $\widehat{\mu}^\star$ by assigning the weights to the maximizer(s) and obtaining $\widehat{u}^\star$ via (15). We summarize the procedure in Algorithm 2, where the terms involving $\eta_i$ with $i = 0$ and $i = 2N + 1$ are set to 0.

In general, $\widehat{F}_X^k(\eta) - \widehat{F}_Y^k(\eta)$ is piece-wise polynomial of degree $(k-1)$, which allows closed-form solutions up to $k = 3$. For $k \geqslant 4$, which is less likely be considered in practice, one can resort to numerical approximations or an optimization treatment by parameterizing $\mu$ with special neural networks, as discussed in Dai et al. (2023).

For further practical consideration, we remark that the extra cost of computing/sampling from the reference $\theta_t$ for the inner loop optimization can be alleviated by instead sampling from the set of samples from previous iterations, in a similar way to experience replay techniques (Lin, 1992) in RL. This eliminates the need of explicitly keeping and periodically updating the reference solution $\theta_t$ and makes the algorithm more streamlined.

## 4 Numerical Experiments

To demonstrate the versatility of our framework, we evaluate LSD on various tasks including supervised learning, reinforcement learning, and portfolio optimization.

### 4.1 Supervised Learning

We examined the performance of LSD on image classification tasks with MNIST and CIFAR-10 datasets. For MNIST, we train a simple 6-layer convolutional neural network for 10 epochs. For CIFAR-10, we use a 20-layer ResNet architecture and train for 200 epochs. In both experiments we set batch size to 128 and adopt stochastic gradient descent (SGD) method to optimize the models, with learning rate set to 0.1 and momentum set to 0.9. We repeat the training procedure on 30 random seeds. The proposed method achieves comparable classification accuracy with SGD method, and more stable cross-entropy loss under $\ell_\infty$-bounded distribution shift, characterized by the average absolute deviation from median (see Theorem 3).

|  | MNIST | | CIFAR-10 | |
| --- | --- | --- | --- | --- |
| Metric | LSD | SGD | LSD | SGD |
| Accuracy | 99.16% | **99.17%** | 91.2% | **91.4%** |
| CE loss | **0.0283** | 0.0289 | **0.293** | 0.339 |
| MAD | **0.0286** | 0.0293 | **0.292** | 0.337 |
| DRO loss ($\rho = 0.1$) derived from the above statistics | **0.0312** | 0.0318 | **0.322** | 0.373 |

Table 1: Test performance measures of LSD versus SGD on MNIST and CIFAR-10.

### 4.2 Reinforcement Learning

We then evaluate the ability of the proposed algorithm of learning risk-averse control policies in two reinforcement learning tasks.

**CliffWalking.** We adopt a modified version of the CliffWalking environment from OpenAI Gym, as illustrated in Figure 2. The action space of the agent is given by $\{0, 1, 2, 3\}$, representing moving by one step in four different directions. When an action is selected, with probability $\epsilon$ the agent will move in a random direction. The agent always start from $s_0$, and receives a reward of $-1$ whenever it falls off the cliffs; reaching the goal instead assigns a positive reward of $+1$. Under both circumstances the episode terminates immediately. Two strategies naturally arise: a risky policy would take the shortest route to the goal and incur a higher chance of falling, while a safe policy prefers taking a detour to minimize the risk.

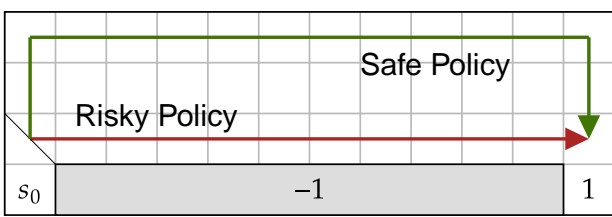

Figure 2: Illustration of the CliffWalking environment.

We tune the values of $\epsilon$ and $\gamma$ to ensure both strategies have similar expected cumulative rewards. We compare the policy learned by LSD (detailed in Appendix B) with the one learned by standard policy gradient method (REINFORCE) (Williams, 1992) under tabular parameterization. The two policies yield similar expected return (0.484 v.s. 0.479), yet our approach achieves consistently lower value of $F^2$, as

demonstrated in Figure 3 (left panel), which demonstrates the risk-averse nature of the learned policy; indeed this can be more evidently observed by examining the density of the return in Figure 3 (right panel), where REINFORCE leads to a higher probability of falling (higher probability mass in negative returns).

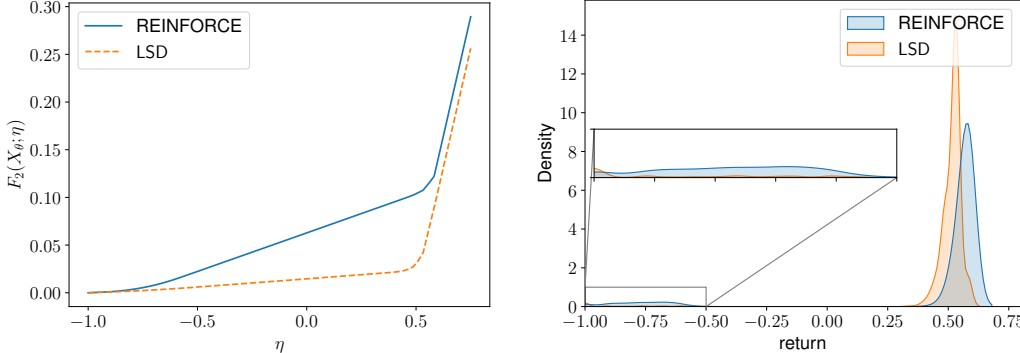

Figure 3: The $F_2$ (left panel) and density (right panel) of the cumulative return in the CliffWalking environment, by executing the policy learned by REINFORCE and LSD-PG, respectively.

**CartPole.** We modify the CartPole-v1 environment from OpenAI gym by perturbing the reward with zero-mean random noise when the cart enters into some certain regions. A risk-aware policy is capable of avoiding such regions without compromising the cumulative return performance. The maximal episode length is set to 500, and the cart position is randomly initialized near the central point. We set the perturb region to $\{x < -0\}$, where $x$ is the cart position. The per-step reward is perturbed by $\delta \sim \text{Uniform}([-\xi, \xi]), \xi = \min(6, 30 \cdot |x|)$ as long as the cart stays in the region. The policy is parameterized by a multi-layer perceptron with two hidden layers of size $(64, 64)$. We average all results over five random seeds. While all methods are able to achieve the optimal expected cumulative return of 500, the policy learned by LSD is able to steer the cart away from the perturbed region, as demonstrated in the right panel of Figure 4. This leads to a more concentrated cumulative return distribution that stochastically dominates that induced by the policy learned by REINFORCE (see the left two panels of Figure 4). Noticeably, CVaR-PG method (Tamar et al., 2015) achieves comparable performance with LSD, but only with a carefully tuned choice of $\alpha$.

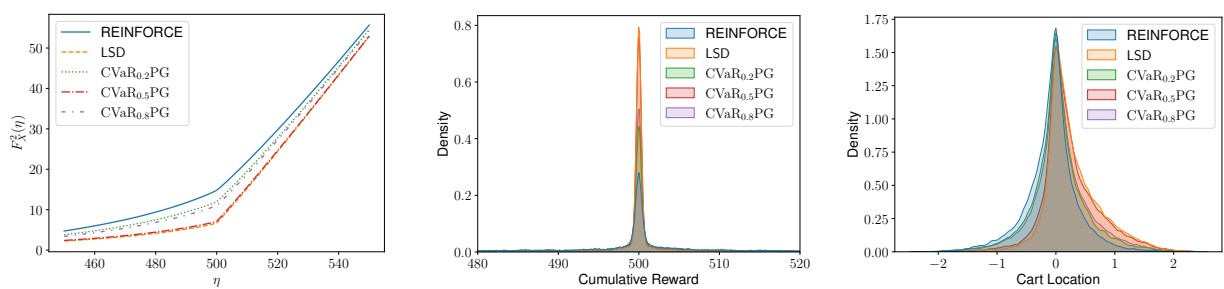

Figure 4: The $F_2$ (left panel) and density (middle panel) of the cumulative return, as well as the density of the visited cart position (right panel) in the CartPole environment, by executing the policy learned by REINFORCE, LSD-PG and $\text{CVaR}_\alpha\text{PG}$, respectively.

### 4.3 Portfolio Optimization

We evaluate the performance of LSD on portfolio optimization with synthesized data and simulate the highly noisy return variables by deploying mixtures of Gaussians with random generated mean and covariance. We set the number of stocks to 100 and the number of Gaussian mixtures to 20. To better reflect the heavy-tailed nature of the problem, we multiply each Gaussian sample's distance to its center by a random multiplier

drawn from $\chi_3^2/3$. Table 1 compares the constructed portfolio with those resulting from the mean-variance approach $MV_\lambda$ (Markowitz & Todd, 2000) using different levels of variance penalty $\lambda$, where Figure 5 further illustrates the density of the portfolio returns.

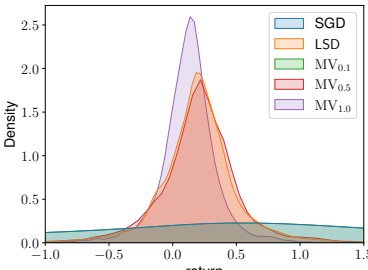

Figure 5: Density of the portfolio returns achieved by different methods.

| Metric | SGD | LSD | $MV_{0.1}$ | $MV_{0.5}$ | $MV_{1.0}$ |
|---|---|---|---|---|---|
| $\mathbb{E}[\text{return}]$ | **0.501** | 0.195 | 0.501 | 0.207 | 0.128 |
| $Var[\text{return}]$ | 7.496 | 0.112 | 7.467 | 0.134 | **0.059** |
| Sharpe ratio | 0.183 | **0.585** | 0.183 | 0.567 | 0.528 |

Table 2: Comparison of LSD versus the mean-variance approach with different variance penalties for portfolio optimization.

While none of the methods simultaneously achieves the highest expected return and the lowest variance, we can evaluate whether the portfolio finds a reasonable trade-off through the Sharpe ratio (Sharpe, 1998), a popular choice for measuring the risk-compensated performance, which distinguishes risky portfolios with catastrophically large variances (highlighted in red in Table 2). We conclude that LSD yields a reasonable construction of portfolio and circumvents the ad-hoc regularization parameter tuning in the mean-risk approach.

## 5 Conclusion

This paper develops the first practical algorithm for finding an optimal solution in terms of (generalized) stochastic dominance for learning and decision making with uncertain outcomes. The method is computationally efficient as it can be easily integrated with existing optimization methods with minimal computational overhead, and come with theoretical guarantees for finite-time convergence. Our work opens up opportunities to further explore the potential of stochastic dominance in risk-averse machine learning applications.

## Acknowledgement

The work of S. Cen and Y. Chi is supported in part by the grants ONR N00014-19-1-2404, NSF DMS-2134080, CCF-2106778 and CNS-2148212. S. Cen is also gratefully supported by Wei Shen and Xuehong Zhang Presidential Fellowship and JP Morgan AI Research Fellowship.

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

## A Connections with DRO

Before finishing up the paper, we demonstrate a connection between SD and DRO, which might be of independent interest. Given $n$ samples $\{x_i\}_{i=1}^n$, the distributionally robust formulation seek to maximize the return under adversarial distribution shifts, i.e.,

$$\inf_{P \ll \widehat{P}_n, P \in \mathcal{B}(\widehat{P}_n)} \mathbb{E}_P[X],$$

where $\widehat{P}_n$ denote the empirical measure of the samples and $\mathcal{B}(\widehat{P}_n)$ is an uncertainty set centering around $\widehat{P}_n$. The following theorem demonstrate that when the uncertainty set is induced by $\ell_\infty$ norm, the objective function can be written in a mean-risk form. The proof is postponed to Appendix E.

**Theorem 3** *It holds that*

$$\inf_{P \ll \widehat{P}_n, \|P - \widehat{P}_n\|_\infty \leqslant \rho/n} \mathbb{E}_P[X] = \mathbb{E}_{\widehat{P}_n}[X] - \rho \text{MAD}_{\widehat{P}_n}[X],$$

*where* $\text{MAD}_{\widehat{P}_n}[X] = \frac{1}{n}\sum_{i=1}^n |x_i - \widetilde{x}|$ *denotes the mean absolute deviation from sample median* $\widetilde{x}$.

It is possible to extend the above result to more general choices of uncertainty set, where the relationship holds asymptotically (similar to Duchi et al. (2021)), from which we refrain for simplicity. On the other hand, we have the following result characterizing the consistency between stochastic dominance and mean-semideviation models (Ogryczak & Ruszczyński, 2001, Theorem 1).

**Theorem 4 (Ogryczak & Ruszczyński (2001))** *Let* $k \geqslant 1$ *and* $X, Y \in \mathcal{L}_k$. *If* $X \succeq_{(k+1)} Y$, *then* $\mathbb{E}[X] \geqslant \mathbb{E}[Y]$ *and*

$$\mathbb{E}[X] - \bar{\delta}_X^{(k)} \geqslant \mathbb{E}[Y] - \bar{\delta}_Y^{(k)}.$$

*Here,* $\bar{\delta}_X^{(k)}$ *denotes the kth central semideviation:*

$$\bar{\delta}_X^{(k)} = \mathbb{E}\left[(\mathbb{E}[X] - X)^k \mathbf{1}_{X \leqslant \mathbb{E}[X]}\right], \quad k = 1, 2, \cdots.$$

In particular, the absolute semideviation at $k = 1$ can be written as

$$\bar{\delta}_X^{(1)} = \int_{-\infty}^{\mathbb{E}[X]} (\mathbb{E}[X] - x)f(x)\mathrm{d}x = \frac{1}{2}\mathbb{E}\left[\left|X - \mathbb{E}[X]\right|\right].$$

Note that it always holds that $\mathbb{E}\left[|X - \widetilde{X}|\right] \leqslant \mathbb{E}\left[|X - \mu_X|\right]$, where $\widetilde{X}$ is the median, and $\mu_X$ is the mean. It follows that when there exists $\theta^\star$ such that $X_{\theta^\star} \succeq_2 X_\theta, \forall \theta \in \Theta$, then $\theta^\star$ can be interpreted as an approximate solution to the robust optimization problem

$$\sup_{\theta \in \Theta} \inf_{P \ll \widehat{P}_n} \left\{\mathbb{E}_P[X_\theta] : \|P - \widehat{P}_n\|_\infty \leqslant \frac{\rho}{n}\right\}$$

for all $\rho \in (0, 1/2)$, in the sense that $\theta^\star$ maximizes a lower bound of the objective function. The approximation error is bounded by $|\mu_X - \widetilde{X}|$.

## B  LSD for Policy Optimization

We detail the procedure of LSD applying to policy optimization in Algorithm 3.

## C  Proof of Proposition 1

Let $\{\theta_t\}_{t=0}^\infty$ be a chain under stochastic dominance rule, i.e., $X_{\theta_i} \succeq_k X_{\theta_j}$ when $i \geqslant j$. The compactness of $\Theta$ assures the existence of a limit point $\widetilde{\theta} \in \Theta$, to which a subsequence of $\{\theta_t\}$ converges. According to the definition of stochastic dominance, for every $\eta \in \mathbb{R}$ the sequence $\left\{F_{X_{\theta_t}}^k(\eta)\right\}_{t=0}^\infty$ is non-decreasing. Since $F_{X_\theta}^k(\eta)$ is continuous with regard to $\theta$, we have

$$F_{X_{\widetilde{\theta}}}^k(\eta) = \lim_{t \to \infty} F_{X_{\theta_t}}^k(\eta).$$

By definition, $X_{\widetilde{\theta}}$ stochastically dominates $X_\theta$ for all $\theta$ from the chain, establishing $\widetilde{\theta}$ as an upper bound of the chain $\{\theta_t\}_{t=0}^\infty$. The existence of maximal element is then guaranteed by Zorn's lemma.

---

**Algorithm 3:** Stochastic Dominance Policy Optimization

---

**1** **Input:** Initialization $\theta_0$.

**2** **for** $t = 0, \cdots, T_{max} - 1$ **do**

**3** $\quad$ Set $\theta_{t,0} = \theta_t$.

**4** $\quad$ **for** $\bar{t} = 0, \cdots, \overline{T}_{max} - 1$ **do**

**5** $\quad\quad$ Sample trajectories $\{\tau_{t,\bar{t},i}\}_{i=1}^N$ with policy $\pi_{\theta_{t,\bar{t}}}$ and $\{\tau_{t,i}\}_{i=1}^N$ with policy $\pi_{\theta_t}$.

**6** $\quad\quad$ Compute cumulative rewards $R_{t,\bar{t},i} = R(\tau_{t,\bar{t},i})$, $R_{t,i} = R(\tau_{t,i})$

**7** $\quad\quad$ Compute $\widehat{u}^\star = \arg\max_{u \in \mathcal{U}_k} \widehat{L}(R_{t,\bar{t}}, R_t, u)$, where

$$\widehat{L}(R_{t,\bar{t}}, R_t, u) := -\frac{1}{N} \sum_{i=1}^N u(R_{t,\bar{t},i}) + \frac{1}{N} \sum_{i=1}^N u(R_{t,i}).$$

**8** $\quad\quad$ Update $\theta$ with

$$\theta_{t,\bar{t}+1} = \theta_{t,\bar{t}} + \frac{\eta_{\bar{t}}}{N} \sum_{i=1}^N \widehat{u}^\star(R_{t,\bar{t},i}) \nabla_\theta \log \pi_{\theta_{t,\bar{t}}}(\tau_{t,\bar{t},i}).$$

**9** $\quad\quad$ **if** $\widehat{\Omega}(R_{t,\bar{t}}, R_t) \leqslant -\epsilon/2$ **then**

**10** $\quad\quad\quad$ Set $\theta_{t+1} = \theta_{t,\bar{t}+1}$.

**11** $\quad\quad\quad$ **Break.**

**12** $\quad$ **if** $\theta_t$ *is not updated* **then**

**13** $\quad\quad$ **Return** $\theta_t$.

---

## D   Proof of Theorem 2

We start by introducing the following lemma which bounds the statistical error due to sampling when $k = 2$.

**Lemma 5** *Let* $\widehat{\Omega}_2(X, Y) = \max_{u \in \mathcal{U}_2} \left\{ -\frac{1}{N} \sum_{i=1}^N u(x_i) + \frac{1}{N} \sum_{i=1}^N u(y_i) \right\}$, *where* $\{x_i, y_i\}_{i=1}^N$ *are i.i.d. samples from* $X$ *and* $Y$. *It holds with probability* $1 - 2\delta'$ *that*

$$\left| \Omega_2(X, Y) - \widehat{\Omega}_2(X, Y) \right| \leqslant \frac{16(|a| + |b|)}{\sqrt{N}} + 6\sqrt{\frac{\log(2/\delta')}{N}}.$$

For notational simplicity, we denote

$$\widehat{\Omega}_2(X_{\theta_{t,\bar{t}}}, X_{\theta_t}) = \max_{u \in \mathcal{U}_2} \left\{ -\frac{1}{N} \sum_{i=1}^N u(x_{\theta_{t,\bar{t}},i}) + \frac{1}{N} \sum_{i=1}^N u(x_{\theta_t,i}) \right\}.$$

By setting $\delta' = \delta/(2T_{\max}(\overline{T}_{\max} + 1))$ in Lemma 5 and invoking the union bound, we have

$$\left| \Omega_2(X_{\theta_{t,\bar{t}}}, X_{\theta_t}) - \widehat{\Omega}_2(X_{\theta_{t,\bar{t}}}, X_{\theta_t}) \right| \leqslant \frac{\epsilon}{4}, \tag{16a}$$

$$\left| \Omega_2(X_{\theta_t^\star}, X_{\theta_t}) - \widehat{\Omega}_2(X_{\theta_t^\star}, X_{\theta_t}) \right| \leqslant \frac{\epsilon}{4}, \tag{16b}$$

for all $0 \leqslant t < T_{\max}, 0 \leqslant \bar{t} < \overline{T}_{\max}$ with probability $1 - \delta$, on which we shall condition in the remaining part of the proof. We remark that with the remaining $\delta$ probability the Algorithm may fail to find a $\theta_t$ qualified for the output condition, or return a sub-optimal solution that accidentally meet the condition. To proceed, we show that $\Omega_k$ satisfies the following triangular inequality:

$$\Omega_k(X, Z) = \max_{\mu \in \Delta'([a,b])} \left\langle F_X^k(\eta) - F_X^k(\eta), \mu(\eta) \right\rangle$$

$$= \max_{\mu \in \Delta'([a,b])} \left\{ \left\langle F_X^k(\eta) - F_Y^k(\eta), \mu(\eta) \right\rangle + \left\langle F_Y^k(\eta) - F_Z^k(\eta), \mu(\eta) \right\rangle \right\}$$

$$\leqslant \max_{\mu \in \Delta'([a,b])} \left\langle F_X^k(\eta) - F_Y^k(\eta), \mu(\eta) \right\rangle + \max_{\mu \in \Delta'([a,b])} \left\langle F_Y^k(\eta) - F_Z^k(\eta), \mu(\eta) \right\rangle$$

$$= \Omega_k(X, Y) + \Omega_k(Y, Z).$$

Let $T$ be the total number of outer iterations. By (16a), for all $0 \leqslant t < T$ we have

$$\Omega_2(X_{\theta_{t+1}}, X_\theta) \leqslant \widehat{\Omega}_2(X_{\theta_{t+1}}, X_\theta) + \frac{\epsilon}{4} \leqslant -\frac{\epsilon}{4}.$$

Denote $\theta_t^\star = \arg \min_\theta \Omega(X_\theta, X_{\theta_t})$. It follows that

$$\Omega_2(X_{\theta_t^\star}, X_{\theta_t}) \leqslant \Omega_2(X_{\theta_T}, X_{\theta_t}) \leqslant \sum_{s=t}^{T-1} \Omega_2(X_{\theta_{s+1}}, X_{\theta_s}) \leqslant -\frac{(T-t)\epsilon}{4}. \tag{17}$$

**Step 1.** We first show that with the choice of $T_{\max}$, Algorithm 1 is guaranteed to return an $\theta_t$. Otherwise, (17) holds for $T = T_{\max}$. On the other hand, we have $0 \leqslant F_X^k(\eta) \leqslant C$ for all $\eta \in [a, b]$. This gives

$$-C \leqslant \Omega_2(X_{\theta_0^\star}, X_{\theta_0}) \leqslant -\frac{T_{\max}\epsilon}{4},$$

or equivalently

$$T_{\max} \leqslant \frac{4C}{\epsilon}.$$

This contradicts with the choice of $T_{\max}$.

**Step 2.** We then prove that the output $\theta_t$ satisfies

$$\Omega_2\left(X_{\theta_t^\star}, X_{\theta_t}\right) \geqslant -\epsilon.$$

According to the update rule, we have

$$\|\theta_{t,\bar{t}+1} - \theta_t^\star\|_2^2 = \|\theta_{t,\bar{t}} - \eta_{\bar{t}} g_{t,\bar{t}} - \theta_t^\star\|_2^2$$

$$= \|\theta_{t,\bar{t}} - \theta_t^\star\|_2^2 - 2\eta_{\bar{t}} \left\langle g_{t,\bar{t}}, \theta_{t,\bar{t}} - \theta_t^\star \right\rangle + \eta_{\bar{t}}^2 \|g_{t,\bar{t}}\|_2^2$$

$$\leqslant \|\theta_{t,\bar{t}} - \theta_t^\star\|_2^2 - 2\eta_{\bar{t}}(\widehat{\Omega}_2(X_{\theta_{t,\bar{t}}}, X_{\theta_t}) - \widehat{\Omega}_2(X_{\theta_t^\star}, X_{\theta_t})) + \eta_{\bar{t}}^2 \|g_{t,\bar{t}}\|_2^2.$$

The last step results from the convexity of $\widehat{\Omega}$. Rearranging terms, we have

$$2\eta_{\bar{t}}\Omega_2(X_{\theta_{t,\bar{t}}}, X_{\theta_t}) - \Omega_2(X_{\theta_t^\star}, X_{\theta_t})$$

$$\leqslant \|\theta_{t,\bar{t}} - \theta_t^\star\|_2^2 - \|\theta_{t,\bar{t}+1} - \theta_t^\star\|_2^2 + \|g_{t,\bar{t}}\|_2^2$$

$$\quad + 2\eta_{\bar{t}}\left[\Omega_2(X_{\theta_{t,\bar{t}}}, X_{\theta_t}) - \widehat{\Omega}_2(X_{\theta_{t,\bar{t}}}, X_{\theta_t})\right] - 2\eta_{\bar{t}}\left[\Omega_2(X_{\theta_t^\star}, X_{\theta_t}) - \widehat{\Omega}_2(X_{\theta_t^\star}, X_{\theta_t})\right]$$

$$\leqslant \|\theta_{t,\bar{t}} - \theta_t^\star\|_2^2 - \|\theta_{t,\bar{t}+1} - \theta_t^\star\|_2^2 + G^2 + \eta_{\bar{t}}\epsilon,$$

where the last step results from (16a) and (16b). Summing over $\bar{t}$, we obtain

$$2\sum_{\bar{t}=1}^{\overline{T}_{\max}} \eta_{\bar{t}}\left[\Omega_2(X_{\theta_{t,\bar{t}}}, X_{\theta_t}) - \Omega_2(X_{\theta_t^\star}, X_{\theta_t})\right] \leqslant \|\theta_t - \theta_t^\star\|_2^2 + G^2 \sum_{\bar{t}=1}^{\overline{T}_{\max}} \eta_{\bar{t}}^2 + \epsilon \sum_{\bar{t}=1}^{\overline{T}_{\max}} \eta_{\bar{t}}. \tag{18}$$

As $\theta_t$ is not updated in the $t$-th outer loop, step 7 of Algorithm 1 ensures

$$\Omega_2(X_{\theta_{t,\bar{t}}}, X_{\theta_t}) \geqslant \widehat{\Omega_2}(X_{\theta_{t,\bar{t}}}, X_{\theta_t}) + \epsilon/4 > -\epsilon/4$$

for $1 \leqslant \bar{t} \leqslant \overline{T}_{\max}$. Combining with the above inequality, we get

$$-\frac{\epsilon}{4} - \Omega_2(X_{\theta_t^\star}, X_{\theta_t}) \leqslant \frac{1}{2\sum_{\bar{t}=1}^{\overline{T}_{\max}} \eta_{\bar{t}}} \left[ \|\theta_t - \theta_t^\star\|_2^2 + G^2 \sum_{\bar{t}=1}^{\overline{T}_{\max}} \eta_{\bar{t}}^2 \right] + \frac{\epsilon}{2}.$$

Note that $\overline{T}_{\max} = \widetilde{\mathcal{O}}(\epsilon^{-2})$ is sufficient to get

$$\frac{1}{2\sum_{\bar{t}=1}^{\overline{T}_{\max}} \eta_{\bar{t}}} \left[ \|\theta_t - \theta_t^\star\|_2^2 + G^2 \sum_{\bar{t}=1}^{\overline{T}_{\max}} \eta_{\bar{t}}^2 \right] \leqslant \frac{\epsilon}{4},$$

which leads to $\Omega_2(X_{\theta_t^\star}, X_{\theta_t}) \geqslant -\epsilon$.

**Step 3.** Finally, we bound the total number of iterations. Let $\overline{T}_t$ be the number of inner iterations in $t-$th outer loop. By (18) we have

$$2\sum_{\bar{t}=1}^{\overline{T}_t-1} \eta_{\bar{t}} \big[ \Omega_2(X_{\theta_{t,\bar{t}}}, X_{\theta_t}) - \Omega_2(X_{\theta_t^\star}, X_{\theta_t}) \big] \leqslant \|\theta_t - \theta_t^\star\|_2^2 + G^2 \sum_{\bar{t}=1}^{\overline{T}_t-1} \eta_{\bar{t}}^2 + \epsilon \sum_{\bar{t}=1}^{\overline{T}_t-1} \eta_{\bar{t}}. \tag{19}$$

Note that $\Omega_2(X_{\theta_{t,\bar{t}}}, X_{\theta_t}) \geqslant \widehat{\Omega_2}(X_{\theta_{t,\bar{t}}}, X_{\theta_t}) + \epsilon/4 > -\epsilon/4$ for $1 \leqslant \bar{t} \leqslant \overline{T}_t - 1$. Combining with the above inequality, we get

$$-\frac{\epsilon}{4} - \Omega_2(X_{\theta_t^\star}, X_{\theta_t}) \leqslant \frac{1}{2\sum_{\bar{t}=1}^{\overline{T}_t-1} \eta_{\bar{t}}} \left[ \|\theta_t - \theta_t^\star\|_2^2 + G^2 \sum_{\bar{t}=1}^{\overline{T}_t-1} \eta_{\bar{t}}^2 \right] + \frac{\epsilon}{2}$$

$$\leqslant \widetilde{\mathcal{O}}\Big(\frac{1}{\sqrt{\overline{T}_t}}\Big) + \frac{\epsilon}{2}.$$

Recall from (17) that $\Omega_2(X_{\theta_t^\star}, X_{\theta_t}) \leqslant -\frac{(T-t)\epsilon}{4}$. We conclude that $\overline{T}_t = \widetilde{\mathcal{O}}\Big(\frac{1}{(T-t)^2\epsilon^2}\Big)$ for $t \leqslant T-3$. Therefore, the total number of iterations is bounded by

$$\sum_{t=0}^{T} \overline{T}_t = \widetilde{\mathcal{O}}\Big( \sum_{t=0}^{T-3} \frac{1}{(T-t)^2\epsilon^2} + 3\overline{T}_{\max} \Big) = \widetilde{\mathcal{O}}(\epsilon^{-2}).$$

### D.1  Proof of Lemma 5

Note that the empirical Rademacher complexity of $\mathcal{U}_2$ with $N$ samples, denoted as $\widehat{\mathfrak{R}}_N(\mathcal{U}_2)$, is the same as that of ReLU functions, i.e.,

$$\widehat{\mathfrak{R}}_N(\mathcal{U}_2) \leqslant \frac{4(|a| + |b|)}{\sqrt{N}}.$$

Therefore, it holds with probability $1 - \delta$ that (Mohri et al., 2018, Theorem 3.3)

$$\Big| \mathbb{E}\big[u_k(X)\big] - \frac{1}{N}\sum_{i=1}^{N} u_k(x_i) \Big| \leqslant 2\widehat{\mathfrak{R}}_N(\mathcal{U}_2) + 3\sqrt{\frac{\log(2/\delta)}{N}}.$$

Similarly,

$$\Big| \mathbb{E}\big[u_k(Y)\big] - \frac{1}{N}\sum_{i=1}^{N} u_k(y_i) \Big| \leqslant 2\widehat{\mathfrak{R}}_N(\mathcal{U}_2) + 3\sqrt{\frac{\log(2/\delta)}{N}}$$

holds with probability $1 - \delta$. By union bound, it holds with probability $1 - 2\delta$ that

$$\Omega_2(X, Y) - \widehat{\Omega}_2(X, Y)$$

$$= \max_{u \in \mathcal{U}_2} \left\{ -\mathbb{E}\big[u(X)\big] + \mathbb{E}\big[u(Y)\big] \right\} - \max_{u \in \mathcal{U}_2} \left\{ -\frac{1}{N} \sum_{i=1}^{N} u(x_i) + \frac{1}{N} \sum_{i=1}^{N} u(y_i) \right\}$$

$$\leqslant \max_{u \in \mathcal{U}_2} \left\{ -\mathbb{E}\big[u(X)\big] + \mathbb{E}\big[u(Y)\big] + \frac{1}{N} \sum_{i=1}^{N} u(x_i) - \frac{1}{N} \sum_{i=1}^{N} u(y_i) \right\}$$

$$\leqslant \frac{16(|a| + |b|)}{\sqrt{N}} + 6\sqrt{\frac{\log(2/\delta)}{N}}.$$

## E  Proof of Theorem 3

The relationship can be established immediately by the following lemma, with $u$ being the distribution shift $P - \widehat{P}_n$.

**Lemma 6** *Let $\tilde{x}$ represents the median of $\{x_i, 1 \leqslant i \leqslant n\}$, and*

$$\mathcal{U} = \left\{ u \in \mathbb{R}^n | \mathbf{1}^\top u = 0, \|u\|_\infty \leqslant \epsilon/n \right\}.$$

*We have*

$$\sup_{u \in \mathcal{U}} u^\top x = \frac{\epsilon}{n} \sum_{i=1}^{n} |x_i - \tilde{x}|.$$

**Proof**  Note that $\mathcal{U}$ is a convex polytope. Therefore $u^\top z$ achieves maximum at one of the vertices of $\mathcal{U}$. Note that the vertices of $\mathcal{U}$ can be written as

$$u_i = \begin{cases} \epsilon/n & i \in \Lambda_+ \\ -\epsilon/n & i \in \Lambda_- \\ 0 & \textbf{otherwise} \end{cases} , \qquad |\Lambda_+| = |\Lambda_-| = \left\lfloor \frac{n}{2} \right\rfloor.$$

When $\Lambda_+$ collects the indices of $\lfloor n/2 \rfloor$ largest values in $\{x_i\}$ and $\Lambda_-$ collecting the smallest values, $u^\top x$ achieves its maximum at

$$u^\top x = \frac{\epsilon}{n} \left[ \sum_{i \in \Lambda_+} x_i - \sum_{i \in \Lambda_-} x_i \right] = \frac{\epsilon}{n} \sum_{i=1}^{n} |x_i - \tilde{x}|.$$

∎

