# OpenReview forum: "Beyond Expectations: Learning with Stochastic Dominance Made Practical"
_TMLR — Accepted by TMLR_

### Review · Reviewer_mLRE · 2025-09-21

**Summary Of Contributions:**

## Summary Of Contributions
- **General framework for risk-aware learning:** This paper tackles the limitation of conventional expectation-based learning by incorporating **stochastic dominance (SD)** as an optimality criterion. SD offers a principled way to compare random outcomes by considering their entire distribution (not just the mean), which is important in high-stakes applications like finance, healthcare, and robotics. However, SD defines only a partial order over distributions, making it non-trivial to use as a training objective. The authors address this by **quantifying the degree of stochastic dominance** as a continuous functional, thereby enabling comparisons between *any* two random variables. This conceptual generalization is a significant contribution that effectively transforms SD from a theoretical preference relation into a usable objective for machine learning models.

- **Novel first-order optimization method (LSD):** Building on the above idea, the paper proposes **Learning with Stochastic Dominance (LSD)**, a new iterative algorithm to find approximately SD-optimal solutions. The method uses a nested-loop optimization: in each outer iteration, it finds a "worst-case" utility function under which the current model is most dominated, and then updates the model to reduce that dominance gap. Importantly, LSD is a **first-order, gradient-based procedure** that can be integrated with standard training (e.g. SGD) with minimal modifications. The authors prove a convergence guarantee: under mild assumptions, LSD will find an $\epsilon$-approximate SD-optimal solution in at most $O(\epsilon^{-2})$ iterations. This theoretical result indicates **minimal computational overhead**, comparable to standard stochastic gradient methods – a strong point in favor of LSD’s practicality.

- **Theoretical guarantees and insights:** The paper provides rigorous analysis to support LSD. It formalizes the SD-optimal solution as a **fixed-point of an optimization process**, ensuring that no other solution stochastically dominates it by more than $\epsilon$. Despite the algorithm’s objective changing at each iteration (non-stationary), the authors establish **finite-time convergence** of LSD to an SD-optimal point. They also draw a connection between their SD-based criterion and **distributionally robust optimization (DRO)**. In particular, they show that optimizing for second-order SD is equivalent to optimizing a surrogate DRO loss: for example, under an $\ell_\infty$ distribution shift, the worst-case expected loss equals the empirical mean loss plus a term proportional to the **mean absolute deviation**. This insight situates LSD within known risk-sensitive frameworks, lending additional intuition and credibility to their approach.

- **Broad empirical evaluation:** The contributions are demonstrated across **diverse learning tasks** – supervised learning, reinforcement learning (RL), and portfolio optimization. In image classification (MNIST and CIFAR-10), LSD achieves essentially the **same accuracy as standard SGD** while yielding a lower and more stable loss under distribution shifts. In RL (CliffWalking and CartPole), LSD learns policies with **similar expected return** to risk-neutral policy gradients, but with significantly reduced risk of catastrophic outcomes. In portfolio optimization, LSD automatically finds a balanced portfolio that avoids extreme variance, achieving a **higher Sharpe ratio** than portfolios from mean-variance trade-offs without needing ad-hoc penalty tuning. These experiments underscore LSD’s versatility and practical impact in **delivering risk-averse solutions without sacrificing performance**. A minor weakness is that comparisons to other established risk-sensitive methods (e.g. CVaR or distributional RL algorithms) are limited, but the provided results strongly support the paper’s main claims.

**Additional Comments:**

## Additional Comments

The paper is generally clear, but the intuition could be strengthened with a simple illustrative example of stochastic dominance before introducing the heavier formalism, which would help orient readers less familiar with the concept. The results convincingly demonstrate LSD’s ability to reduce risk without sacrificing expected performance, with the reinforcement learning experiments providing particularly strong evidence of this balance. A further practical advantage worth highlighting is that LSD eliminates the need for manual tuning of risk trade-off parameters, setting it apart from many mean-variance or DRO-based methods. Finally, future research directions could include extending LSD to fairness-sensitive machine learning, scaling it to large deep learning models, or tailoring it to specific high-stakes domains such as finance or healthcare, where risk considerations are central.

**Audience:**

Yes

**Audience Explanation:**

Individuals in TMLR's audience would be interested in this work for the following reasons:

- **Risk-sensitive learning is important:** The work tackles a key issue in ML – reliability under uncertainty – that spans multiple application areas.

- **Novel methodology with practical impact:** LSD is both theoretically interesting and practically implementable in standard training pipelines.

- **Cross-domain appeal:** Experiments across vision, RL, and finance ensure the paper is relevant to multiple sub-communities.

- **Theoretical and conceptual interest:** Bridges stochastic dominance from economics/decision theory to modern ML and DRO, sparking potential new research directions.

**Broader Impact Concerns:**

## Broader Impact Concerns

- The paper lacks an explicit broader impact section.
- Positive impacts: safer AI systems in finance, healthcare, or robotics by reducing catastrophic risk.
- Risks: overly conservative decisions might lower utility or unfairly disadvantage certain groups.
- Authors should reflect on both sides and encourage careful application of LSD in real-world deployments.

**Claims And Evidence:**

Yes

**Claims Explanation:**

- *Claim 1: “First general framework for learning with stochastic dominance.”* – Supported by a survey of prior work showing earlier SD methods were theoretical or impractical. The authors provide a concrete algorithm (LSD) with proofs of convergence and existence of SD-optimal solutions. This fills a genuine gap in the literature with a practical solution.

- *Claim 2: “LSD has minimal overhead and converges to an $\epsilon$-optimal solution.”* – Theoretical complexity analysis proves convergence in $O(\epsilon^{-2})$ iterations, matching SGD. Algorithm description and proofs provide confidence that this claim is accurate.

- *Claim 3: “LSD achieves comparable performance to risk-neutral methods while improving risk profiles.”* – Empirical results (MNIST, CIFAR-10, RL, portfolio optimization) confirm equal expected performance but reduced variance, improved tail outcomes, and higher Sharpe ratios. This directly backs the claim.

- *Claim 4: “Broad applicability and versatility.”* – Demonstrated through experiments across classification, RL, and finance. Although more baselines would be useful, the results clearly support the claim of generality.

**Requested Changes:**

## Requested Changes

- **(Critical) Include comparisons with other risk-aware baselines:**
  While the experiments demonstrate LSD’s ability to maintain expected performance while improving risk profiles, the empirical evaluation would be much stronger with direct comparisons to **established risk-sensitive methods**. For example, in supervised learning, including a baseline such as CVaR (Conditional Value at Risk) optimization or a distributionally robust training objective (e.g., adversarial DRO or worst-case risk minimization) would show how LSD performs relative to methods already in practice. In reinforcement learning, comparing against risk-aware RL algorithms (such as CVaR Policy Gradient or distributional RL methods like C51 or QR-DQN) would contextualize LSD’s gains in reducing catastrophic outcomes. Without these baselines, it is difficult to assess whether LSD truly advances the state of the art or simply achieves what existing methods can already provide. Adding these comparisons is essential to establish LSD as a genuinely superior approach rather than just an alternative formulation.

- **(Critical) Clarify the role of different dominance orders:**
  The paper introduces a general framework for $k$-th order stochastic dominance, but in practice, all experiments focus on second-order dominance (which aligns with risk-averse decision-making and variance sensitivity). It is unclear whether LSD is equally effective or even tractable for higher orders, such as third-order dominance (capturing skewness preferences) or beyond. The authors should explicitly state whether higher-order extensions are supported, whether the theory and algorithm remain valid, and if so, whether they have practical significance. If the focus is mainly on $k=2$, this limitation should be acknowledged clearly to avoid overselling the generality of the contribution. Alternatively, even a small-scale demonstration of $k=3$ on synthetic data would reassure readers that the framework is not limited to just second-order dominance.

- **(Critical) Discuss limitations and edge cases:**
  Currently, the paper emphasizes the strengths of LSD but provides little discussion of its limitations. It would improve transparency and academic rigor to explicitly acknowledge edge cases. For instance, in hypothesis spaces with multiple **non-dominated solutions**, LSD may converge to one arbitrary solution without guarantees of uniqueness; readers should know whether this affects interpretability or robustness. Similarly, the assumptions underlying the theoretical results, such as compactness of the hypothesis space and continuity of loss distributions, may not hold in high-dimensional deep learning settings. The authors should comment on how violations of these assumptions might impact LSD’s performance. Finally, discussion of computational scaling in very large models would be useful, since even small overheads in the inner–outer loop structure could become significant. Highlighting these points will strengthen the credibility of the work by showing awareness of its boundaries.

- **(Suggested) Add a Broader Impact discussion:**
  The submission would benefit from a dedicated section reflecting on the societal and ethical implications of risk-sensitive learning. On the positive side, LSD could reduce the frequency of catastrophic failures in applications like healthcare, finance, or autonomous systems, where tail risks are critical. On the other hand, excessive risk aversion could result in **overly conservative models**, potentially harming efficiency or fairness (e.g., consistently rejecting high-risk/high-reward loan applicants, or under-treating patients who could benefit from riskier but more effective therapies). The authors should also mention possible misuse, such as financial institutions employing LSD-based models in ways that unfairly disadvantage certain groups. Including this discussion would not only meet TMLR expectations but also demonstrate responsible research practice.

- **(Suggested) Minor clarity and presentation improvements:**
  While the paper is generally well-written, several improvements could make it more accessible. First, an **illustrative toy example** early in the paper (e.g., comparing two distributions with simple means and variances) would help readers unfamiliar with stochastic dominance understand the intuition behind the dominance functional before diving into heavy formalism. Also, a careful proofreading pass to fix minor grammatical issues and streamline exposition would enhance readability. These are not critical changes, but they would improve the presentation.

---

> ### Author Response · Authors · 2025-10-06
> **Response to Reviewer mLRE [1/2]**
>
> ### **Requested Changes**
> > 1. **(Critical) Include comparisons with other risk-aware baselines**: While the experiments demonstrate LSD’s ability to maintain expected performance while improving risk profiles, the empirical evaluation would be much stronger with direct comparisons to established risk-sensitive methods. For example, ...
>
> We thank the reviewer for raising this point. Please see our common response on the comparison with CVaR baselines.
>
> ---
>
> > 2. **(Critical) Clarify the role of different dominance orders**: The paper introduces a general framework for $k$-th order stochastic dominance, but in practice, all experiments focus on second-order dominance (which aligns with risk-averse decision-making and variance sensitivity). It is unclear whether LSD is equally effective or even tractable for higher orders, such as third-order dominance (capturing skewness preferences) or beyond. The authors should explicitly state whether higher-order extensions are supported, whether the theory and algorithm remain valid, and if so, whether they have practical significance. If the focus is mainly on $k=2$, this limitation should be acknowledged clearly to avoid overselling the generality of the contribution. Alternatively, even a small-scale demonstration of $k=3$ on synthetic data would reassure readers that the framework is not limited to just second-order dominance.
>
> We appreciate the reviewer's feedback and would like to clarify that we have included discussions on $k>2$ for both theoretical analysis and practical implementation. In Theorem 2's remark, we explicitly stated that:
> > While we state the theorem for $k=2$ for simplicity, the analysis can be easily generalized to $k>2$ by adopting the Rademacher complexity of $\mathcal{U}_k$ and upper bound of $F^k$ accordingly.
>
> We have also discussed algorithm implementation with $k>2$ in Section 3.3 "Practical Implementation":
> > In general, $\widehat{F}_X^{k}(\eta) - \widehat{F}_Y^{k}(\eta)$ is piece-wise polynomial of degree $(k-1)$, which allows closed-form solutions up to $k=3$. For $k\ge 4$, which is less likely be considered in practice, one can resort to numerical approximations or an optimization treatment by parameterizing $\mu$ with special neural networks, as discussed in Dai et al. (2023).
>
> ---
>
> > 3. **(Critical) Discuss limitations and edge cases**: Currently, the paper emphasizes the strengths of LSD but provides little discussion of its limitations. It would improve transparency and academic rigor to explicitly acknowledge edge cases. For instance, in hypothesis spaces with multiple non-dominated solutions, LSD may converge to one arbitrary solution without guarantees of uniqueness; readers should know whether this affects interpretability or robustness. Similarly, the assumptions underlying the theoretical results, such as compactness of the hypothesis space and continuity of loss distributions, may not hold in high-dimensional deep learning settings. The authors should comment on how violations of these assumptions might impact LSD’s performance. Finally, discussion of computational scaling in very large models would be useful, since even small overheads in the inner–outer loop structure could become significant. Highlighting these points will strengthen the credibility of the work by showing awareness of its boundaries.
>
> We thank the reviewer for the suggestion.
> - **uniqueness of algorithm outcome**: Please see our common response on non-uniqueness and interpretation of the outcome.
> - **theoretical assumptions may not hold in high-dimensional deep learning settings**: The assumptions in Proposition 1 generally hold as long as $X_\theta$ are continuous random variables and that the samples $x_i$ are continuous to $\theta$. Generalizing Theorem 2 to high-dimensional deep learning settings is non-trivial, and we remark that optimization theory for deep learning has achieved limited success beyond a few simplified settings with heavy assumptions, e.g., studying the model's linearization around its initial point. Addressing this would be beyond the scope of this work.
> - **computational scaling:** We discussed in section 3.3 on the space and time complexity for computing $\widehat{u}^\star$.
> - **overhead of inner-outer loop:** We discussed in section 3.3 on how one can modify the algorithm to meet practical consideration:
> > For further practical consideration, we remark that the extra cost of computing/sampling from the reference $\theta_t$ for the inner loop optimization can be alleviated by instead sampling from the set of samples from previous iterations, in a similar way to experience replay techniques (Lin, 1992) in RL. This eliminates the need of explicitly keeping and periodically updating the reference solution $\theta_t$ and makes the algorithm more streamlined.
>
> ---

---

> ### Author Response · Authors · 2025-10-06
> **Response to Reviewer mLRE [2/2]**
>
> > 4. **(Suggested) Minor clarity and presentation improvements:** While the paper is generally well-written, several improvements could make it more accessible. First, an illustrative toy example early in the paper (e.g., comparing two distributions with simple means and variances) would help readers unfamiliar with stochastic dominance understand the intuition behind the dominance functional before diving into heavy formalism. Also, a careful proofreading pass to fix minor grammatical issues and streamline exposition would enhance readability. These are not critical changes, but they would improve the presentation.
>
> We thank the reviewer for the suggestion. We illustrate in Figure 1 how a more concentrated distribution ($\mathcal{N}(0,1)$) yields a consistently lower $F^2$ curve than a more dispersed one ($\mathcal{N}(0,2)$), thereby demonstrating second-order stochastic dominance. We have also corrected minor grammatical and notational issues in the revision.
>
> ---
>
> #### Broader Impact Concerns
>
> > - **(Suggested) Add a Broader Impact discussion:** The submission would benefit from a dedicated section reflecting on the societal and ethical implications of risk-sensitive learning. On the positive side, LSD could reduce the frequency of catastrophic failures in applications like healthcare, finance, or autonomous systems, where tail risks are critical. On the other hand, excessive risk aversion could result in overly conservative models, potentially harming efficiency or fairness (e.g., consistently rejecting high-risk/high-reward loan applicants, or under-treating patients who could benefit from riskier but more effective therapies). The authors should also mention possible misuse, such as financial institutions employing LSD-based models in ways that unfairly disadvantage certain groups. Including this discussion would not only meet TMLR expectations but also demonstrate responsible research practice.
> >  - Positive impacts: safer AI systems in finance, healthcare, or robotics by reducing catastrophic risk.
> >  - Risks: overly conservative decisions might lower utility or unfairly disadvantage certain groups.
> >  - Authors should reflect on both sides and encourage careful application of LSD in real-world deployments.
>
> We appreciate the reviewer’s thoughtful suggestion. We agree that a broader impact discussion around risk-sensitive learning is important. However, as this work primarily introduces a preliminary framework for incorporating stochastic dominance into machine learning, we view an in-depth societal and ethical analysis as beyond the intended scope of this paper. Risk aversion frameworks, including stochastic dominance, have been extensively studied for decades, and their societal implications have been discussed in various contexts. We see our contribution as laying the technical foundation, and we believe a dedicated broader impact discussion would be more meaningful once concrete applications of our framework are developed.

---

### Review · Reviewer_Y1PH · 2025-09-21

**Summary Of Contributions:**

The authors present a comprehensive framework based on stochastic dominance to solve optimisation problems under uncertainty. They first motivate the method reasoning about how stochastic dominance provides a naturally risk-aware optimisation objective without the need for trade-offs or multiple objective formulations. They analyse the main caveat of using SD as an optimisation objective; the fact that SD only provides a partial ordering of variables, and does not guarantee a global optimum, and propose a solution by relaxing the criteria into a scalar formulation of a stochastic dominance gap. They provide an algorithmic approach to learning with SD, theoretical analysis of convergence properties and some empirical results demonstrating the intuition behind SD learning solutions obtained in standard classification, RL and decision making problems. Overall I find it a good paper with interesting insights.
## Strengths
- The paper introduces a seemingly practical method to learn with SD and provides both theoretical and empirical results.
- The authors cover comprehensively the intuition behind optimizing the proposed generalised SD functional.
- The paper is well structured and easy to follow.

## Weaknesses
- Some sections or concepts are missing detail, see below for questions.
- The practical utility of learning with SD is unclear to me yet, but I trust that it could be a powerful approach in some problems.

**Audience:**

Yes

**Audience Explanation:**

Yes. The work proposes a general-purpose approach to risk-aware learning based in stochastic dominance, relevant to learning theory, robust optimization, risk-sensitive RL/decision making and to other field-specific (e.g. finance) mean–risk trade-off problems. The paper connects SD to utility theory and DRO, which will be useful for readers working on principled robustness and distributional optimization approaches. The empirical section shows the proposed method is practical across standard ML settings.

**Broader Impact Concerns:**

No concerns.

**Claims And Evidence:**

Yes

**Claims Explanation:**

Yes, the authors provide justification, results and evidence for the claims in the paper. There are no unsupported claims.

**Requested Changes:**

I have some questions and comments that I would appreciate the authors clarifying.
- There is not really a discussion or analysis on the particular choice of the intervals $[a,b]$, particularly in how it influences the generalised SD functional or how the choice of bounds would affect learning dynamics. Some comment on this would be useful.
- There is some discussion on how SD with $k=2$ is somehow equivalent to the CVaR, but there is no specific comparison in the empirical examples against CVaR optimisation methods (very prevalent in e.g. RL [1]). Could the authors comment on specific advantages of SD versus CVaR approaches?
- The authors mention that existing risk-aware optimisation frameworks (mean-risk, CVaR) often require trade-off design or tuning of relative weights and these decisions are not usually formally justified. I agree, but because of this added design choice, these methods also provide a way to control the risk aversion induced in the optimal solutions obtained. Would there be an analogous way of tuning risk-aversion in favour of  e.g. higher cumulated probability mass in higher utility regions of the space?
- There is plenty of discussion on using first and second order distribution functions, but it is not clear to me under what circumstances one would want to use higher order $k$, and what would that mean intuitively for the risk aversion of the solutions obtained.
- I think the paper would benefit of having a general, abstracted subsection where Learning with SD is formally introduced in an objective agnostic way *before* the examples provided (i.e. the examples in pg 5 were very clear and helped ground the SD approach to specific optimisation problems but it would be clearer if an abstract, formal definition of the framework would be presented before, and then particularised to each example).


[1] Chow, Yinlam, et al. "Risk-sensitive and robust decision-making: a cvar optimization approach." Advances in neural information processing systems 28 (2015).

---

> ### Author Response · Authors · 2025-10-06
> **Response to Reviewer Y1PH [1/2]**
>
> ### **Weaknesses**
>
> > 1. Some sections or concepts are missing detail, see below for questions.
>
> We appreciate the reviewer's feedback and we have made changes accordingly in the revision.
>
> ---
>
> > 2. The practical utility of learning with SD is unclear to me yet, but I trust that it could be a powerful approach in some problems.
>
> We thank the reviewer for this perspective. While we acknowledge that the full range of applications may not yet be clear, we view this work as an initial step toward integrating stochastic dominance with various learning tasks and hope it will inspire further applications.
>
> ---
>
> ### **Requested Changes**
> > 1. There is not really a discussion or analysis on the particular choice of the intervals [a, b], particularly in how it influences the generalised SD functional or how the choice of bounds would affect learning dynamics. Some comment on this would be useful.
>
> We thank the reviewer for pointing this out and we are happy to briefly discuss how samples outside the range are treated during learning. Specifically, when a sample $x_{t,\bar{t},i}$ lies in $(b,\infty)$, it is effectively ignored by the algorithm, as the gradient contribution is zero. When it falls in $(-\infty,a)$, the gradient weight is determined by $\widehat{u}^\star$ in line 6 of Algorithm 1. For example, in the case $k=2$, Algorithm 2 gives $\widehat{u}\_1(\eta) = -1$ for $\eta < a$, which leads to a gradient weight $\partial_{x_{t,\bar{t},i}} \widehat{u}^\star(x_{t,\bar{t},i}) = 1$. Thus, for $k=2$, LSD reduces to standard SGD for samples below $a$.
> Please also see our common response on relaxation to [a, b].
>
> ---
>
> > 2. There is some discussion on how SD with $k=2$ is somehow equivalent to the CVaR, but there is no specific comparison in the empirical examples against CVaR optimisation methods (very prevalent in e.g. RL [1]). Could the authors comment on specific advantages of SD versus CVaR approaches?
>
> We thank the reviewer for raising this point. Please see our common response on the comparison with other baselines.
>
> ---
>
> > 3. The authors mention that existing risk-aware optimisation frameworks (mean-risk, CVaR) often require trade-off design or tuning of relative weights and these decisions are not usually formally justified. I agree, but because of this added design choice, these methods also provide a way to control the risk aversion induced in the optimal solutions obtained. Would there be an analogous way of tuning risk-aversion in favour of e.g. higher cumulated probability mass in higher utility regions of the space?
>
> We thank the reviewer for the feedback. Achieving higher cumulated probability mass in some high utility regions $[\eta_{high}, \infty)$ amounts to minimize $F(\eta_{high})$. This may contradict with the stochastic dominance goal, as $\mathcal{N}(\eta_{high}-1; 0.1)$ stochastically dominates $\mathcal{N}(\eta_{high}-1; 100)$ in second order but the former has higher $F(\eta_{high})$. One can incoporate an additional factor
> $$
> \lambda \cdot \mathbf{1}\_{x_{t,\bar{t},i}\le \eta_{high}},
> $$
> with strength parameter $\lambda > 0$ in the sample weight of Algorithm 1 to force the preference.
> Please also see our common response on non-uniqueness and interpretation of the outcome.

---

> ### Author Response · Authors · 2025-10-06
> **Response to Reviewer Y1PH [2/2]**
>
> > 4. There is plenty of discussion on using first and second order distribution functions, but it is not clear to me under what circumstances one would want to use higher order $k$, and what would that mean intuitively for the risk aversion of the solutions obtained.
>
> We thank the reviewer for raising this point. Moving from first to higher orders of stochastic dominance in general corresponds to incorporating sensitivity to higher moments such as skewness (third order) or even kurtosis and beyond. Intuitively, higher-order dominance captures increasingly fine-grained forms of risk aversion: for example, third-order stochastic dominance (TSD) reflects a preference for positively skewed distributions (avoiding downside tail risks while favoring upside potential) [1].
>
> [1] Chan, Raymond H., Ephraim Clark, Xu Guo, and Wing-Keung Wong. "New development on the third-order stochastic dominance for risk-averse and risk-seeking investors with application in risk management." Risk Management 22, no. 2 (2020): 108-132.
>
> ---
>
> > 5. I think the paper would benefit of having a general, abstracted subsection where Learning with SD is formally introduced in an objective agnostic way before the examples provided (i.e. the examples in pg 5 were very clear and helped ground the SD approach to specific optimisation problems but it would be clearer if an abstract, formal definition of the framework would be presented before, and then particularised to each example).
>
> We thank the reviewer for the suggestion. We agree that an abstract, objective-agnostic formulation of learning with SD could be valuable for clarity. However, providing a full formal definition requires several paragraphs of setup including clarifying the notion of optimality in SD learning, which we found would risk delaying the main narrative.

---

> > ### Comment · Reviewer_Y1PH · 2025-10-19
> >
> > I thank the authors for the detailed rebuttal replies. I consider my points addressed and I am quite satisfied with the changes and additions to the paper.

---

> > > ### Author Response · Authors · 2025-10-21
> > >
> > > Thank you so much! We are happy to hear you find the response satisfactory.

---

### Review · Reviewer_6wf5 · 2025-09-22

**Summary Of Contributions:**

The article proposes establishing stochastic dominance as a risk-sensitive alternative to expected utility for comparing random variables.
While this is already standard practice in some fields (e.g., finance, decision theory), such an approach is not widely used in machine learning.
However, the authors show that important areas of machine learning, such as supervised learning or reinforcement learning, can also be
seamlessly embedded in a stochastic dominance framework.

The article names two reasons that (in the authors' opinion) prevent the further spread of SD in machine learning: (1) SD leaves certain variables incomparable and thus
does not correspond to the optimization mindset that prevails in ML, and (2) SD requires the comparison of variables across a continuum and is
therefore difficult to evaluate.

The article intends to address both of the challenges (1) and (2) mentioned above: For (1), a generalized stochastic dominance criterion is proposed,
which should enable a comparison of any variables. For (2), a first-order algorithm is proposed, which makes it possible to efficiently calculate an
(approximately) optimal solution according to the proposed generalized SD criterion. The article demonstrates the applicability of the proposed framework
using numerical examples from the fields of SL, RL, and portfolio optimization.


Strenghts:

- The paper addresses a relevant and interesting topic: Introducing SD-based Learning in ML seems to be a very good idea, as this allows for addressing
several shortcomings of the traditional focus on the expected utility / expectation only. The concrete shortcoming mentioned here is its lack of risk-sensitivity,
but in my opinion the argument extends to nonstandard scaled data and other interesting areas.

- The possibility of efficient algorithmic verification of SD seems very attractive to me. Progress in this direction has the potential to make SD methods, which
appear very attractive in theory, relevant for practical work as well.

- The possibility of treating seemingly different areas such as supervised learning, reinforcement learning, and portfolio optimization within the same formal
framework strikes me as very elegant. It also reinforces the relevance for various fields of application.


Weaknesses:

- It is not really clear (to me) to what extent the concept of stochastic dominance is generalized here (as is claimed). As I understand it, the main proposal
is to search for an undominated solution (which always exists under mild conditions, see Prop. 1) instead of a best solution in the sense of SD (which generally
does not exist). To my knowledge, this approach is fairly standard in the context of SD (and non-total orders in general). However, this comment refers to the
conceptual innovation and not to the proposed algorithm.

- I find it difficult to identify the main contribution of the paper. While it is clear to me that an algorithm for more efficient discovery (compared to SoTA methods) of SD-undominated acts is proposed, the generality of the main theorem is unclear to me: Does this refer to SSD (as mentioned in line 1), or is an undominated act of order k actually found, as suggested by the last line of the theorem? I would appreciate some clarification on this point.

- My biggest concerns relate to the interpretation of the “generalization” proposed here: While ERM solutions are unique (in many relevant cases), this is by no means true for undominated SD solutions. So if one algorithm (e.g., in an SL context) finds an undominated solution X and another finds an undominated solution X' (in our example, different forecasting algorithms), these will generally make very different predictions. The number of undetermined models is often very large, which means that the model found is, to a certain extent (except for certain worst-case guarantees), arbitrary. Shouldn't we rather try to sensibly strengthen the ordering power of SD methods instead of simply going with an underdetermined model?

**Audience:**

Yes

**Audience Explanation:**

I think incorporating SD in learning problems is a promising line of research that is often hindered by the complexity associated with checking for SD. In this sense, I think any step towards making SD more accessible for practitioners (i.e., more efficient to check) is very much needed and relevant.

**Broader Impact Concerns:**

No broader impact concerns.

**Claims And Evidence:**

Yes

**Claims Explanation:**

The main result, Theorem 2, is supported by a detailed mathematical proof.

The proposed algorithm is demonstrated along three different applications.

**Requested Changes:**

- At several places it is claimed that SD is a partial order, i.e., antisymmetric. However, afaik, this is not the case:
Two variables may be equivalent with respect to SD, without being equal (just think of X,Y following the same distribution under FSD).

- If I understand correctly, the “generalization” of stochastic dominance (compare Equation (6)) consists of checking whether (strict)
stochastic dominance in the reverse direction does not exist. In other words, X dominates Y in the generalized sense if Y does not SD-dominate
X strictly.  It is therefore not entirely clear to me to what extent this is actually a conceptual generalization.

- In connection with this, I am also surprised by the wording in Proposition 1: “... THE non-dominated solution ...”. In general, there will be
quite a few undominated variables. Thus, there will generally be no unique solution. Please reformulate this appropriately.

- When discussing related work on generalizing stochastic dominance, the following recent attempts to do so seem to be missing. Both papers follow
the idea of strengthening the SD-order to increase comparability among variables, which seems a somewhat similar goal to what is pursued here.

- https://proceedings.neurips.cc/paper_files/paper/2024/hash/b1f140eeee243db24e9e006481b91cf1-Abstract-Conference.html

- https://jmlr.org/papers/v24/22-0902.html

- In Equation (5), how do you choose the interval boundaries a and b in practice? Cutting off the tails of the distribution functions might
imply that the undominated variables above the interval are not necessarily undominated on the entire real line. Thus, this restriction may
result in dominated variables being selected, correct? If so, the choice of a and b should be well justified. It would also be nice to learn
a little more about the “technical subtleties” mentioned (see paragraph under equation (5)).

- I think there is a chnage in notation when comparing Equations (1),(2),(3) and Figure 1 and the paragraph above it: While in the equations the kth order
distribution is donoted by F^k, in the figure and its description it is denoted by F_k.

- I find the explanations surrounding equation (6) somewhat confusing:  The equation itself specifies the condition that X_{\theta^*} is not SD-dominated by X_{\theta}.
Below this, it is stated that X_{\theta^*} is not dominated by any other variable. It seems to me that an all-quantifier is missing before equation (6) (\forall \theta).
Only then does the reformulation at the beginning of Section 3.1 make sense.

- The idea of the relaxation in Equation (7) seems to be close to the notion of almost SD, just replaced by almost non-SD. While I think the idea is nice in
principle and can understand it from an optimization standpoint, I still wonder how strong the statement is in the end: Especially when combined with the truncated
tails from equation (5), it seems to me that a lot of precision is lost (especially since Theorem 2 also only provides an almost undominated solution).
A little discussion on this would certainly be enlightening.

- The reformulation of condition (6) at the beginning of Section 3.1 seems a little sloppy to me. I understand that a maximum element $X_{\theta^*}$ can be characterized equivalently by requiring that

$
\min_{\theta \in \Theta} \, \Omega(X_{\theta}, X_{\theta^*})  \geq  0 .
$

However, no requirement is made for the $\arg\min$ specified here. Perhaps one could describe in more detail that this $\arg\min$ is a candidate for an undominated solution. If one starts with a dominated $X_{\theta^*}$, one still finds an $\arg\min$, but this does not satisfy the condition.

- I believe the symbol x_theta (lowercase x), which is first used in the bottom paragraph on page 7, was never formally defined.

---

> ### Author Response · Authors · 2025-10-06
> **Response to Reviewer 6wf5 [1/2]**
>
> ### **Weaknesses**
>
> > 1. It is not really clear (to me) to what extent the concept of stochastic dominance is generalized here (as is claimed). As I understand it, the main proposal is to search for an undominated solution (which always exists under mild conditions, see Prop. 1) instead of a best solution in the sense of SD (which generally does not exist). To my knowledge, this approach is fairly standard in the context of SD (and non-total orders in general). However, this comment refers to the conceptual innovation and not to the proposed algorithm.
>
> We thank the reviewer for this comment. Our notion of generalized stochastic dominance is motivated by the need for a concrete numerical criterion to guide optimization, whereas standard stochastic dominance only provides a ternary relation (dominates / dominated / incomparable). To our knowledge, this specific formulation has not been explored in the context of stochastic dominance optimization. That said, given its natural derivation, we would not be surprised if related ideas have appeared elsewhere, and we would greatly appreciate any pointers to such work.
>
> ---
>
> > 2. I find it difficult to identify the main contribution of the paper. While it is clear to me that an algorithm for more efficient discovery (compared to SoTA methods) of SD-undominated acts is proposed, the generality of the main theorem is unclear to me: Does this refer to SSD (as mentioned in line 1), or is an undominated act of order k actually found, as suggested by the last line of the theorem? I would appreciate some clarification on this point.
>
> We appreciate reviewer's feedback and would like to clarify that we have included discussions on $k>2$ for both theoretical analysis and practical implementation. In Theorem 2's remark, we explicitly stated that:
> > While we state the theorem for $k=2$ for simplicity, the analysis can be easily generalized to $k>2$ by adopting the Rademacher complexity of $\mathcal{U}_k$ and upper bound of $F^k$ accordingly.
>
> We have also discussed algorithm implementation with $k>2$ in Section 3.3 "Practical Implementation":
> > In general, $\widehat{F}_X^{k}(\eta) - \widehat{F}_Y^{k}(\eta)$ is piece-wise polynomial of degree $(k-1)$, which allows closed-form solutions up to $k=3$. For $k\ge 4$, which is less likely be considered in practice, one can resort to numerical approximations or an optimization treatment by parameterizing $\mu$ with special neural networks, as discussed in Dai et al. (2023).
>
> ---
>
> > 3. My biggest concerns relate to the interpretation of the “generalization” proposed here: While ERM solutions are unique (in many relevant cases), this is by no means true for undominated SD solutions. So if one algorithm (e.g., in an SL context) finds an undominated solution X and another finds an undominated solution X' (in our example, different forecasting algorithms), these will generally make very different predictions. The number of undetermined models is often very large, which means that the model found is, to a certain extent (except for certain worst-case guarantees), arbitrary. Shouldn't we rather try to sensibly strengthen the ordering power of SD methods instead of simply going with an underdetermined model?
>
> We thank the reviewer's feedback. Please see our common response on non-uniqueness and interpretation of the outcome.
>
> ---
>
> ### **Requested changes**
> > 1. At several places it is claimed that SD is a partial order, i.e., antisymmetric. However, afaik, this is not the case: Two variables may be equivalent with respect to SD, without being equal (just think of X,Y following the same distribution under FSD).
>
> Stochastic dominance defines a partial order on distributions (or equivalently, equivalence classes of random variables up to law), not on the raw measurable functions themselves. With this standard convention, antisymmetry is satisfied: if $X \succeq_k Y$ and $Y \succeq_k X$, then X and Y are equal in distribution, i.e., they are the same element of the partially ordered set. In revision we have corrected the misleading statement to emphasize that SD is a partial order on distributions.
>
> ---
>
> > 2. If I understand correctly, the “generalization” of stochastic dominance (compare Equation (6)) consists of checking whether (strict) stochastic dominance in the reverse direction does not exist. In other words, X dominates Y in the generalized sense if Y does not SD-dominate X strictly. It is therefore not entirely clear to me to what extent this is actually a conceptual generalization.
>
> We thank the reviewer for raising this point. Please see our response to Weakness 1.

---

> ### Author Response · Authors · 2025-10-06
> **Response to Reviewer 6wf5 [2/2]**
>
> > 3. In connection with this, I am also surprised by the wording in Proposition 1: “... THE non-dominated solution ...”. In general, there will be quite a few undominated variables. Thus, there will generally be no unique solution. Please reformulate this appropriately.
>
> Thanks for brining this to our attention. We have rephrased it to "a non-dominated solution" in our revision.
>
> ---
>
> > 4. When discussing related work on generalizing stochastic dominance, the following recent attempts to do so seem to be missing. Both papers follow the idea of strengthening the SD-order to increase comparability among variables, which seems a somewhat similar goal to what is pursued here.
>
> > https://proceedings.neurips.cc/paper_files/paper/2024/hash/b1f140eeee243db24e9e006481b91cf1-Abstract-Conference.html
>
> > https://jmlr.org/papers/v24/22-0902.html
>
> We thank the reviewer for the missing reference. We have included those references in our revision.
>
> ---
>
> > 5. In Equation (5), how do you choose the interval boundaries a and b in practice? Cutting off the tails of the distribution functions might imply that the undominated variables above the interval are not necessarily undominated on the entire real line. Thus, this restriction may result in dominated variables being selected, correct? If so, the choice of a and b should be well justified. It would also be nice to learn a little more about the “technical subtleties” mentioned (see paragraph under equation (5)).
>
> We thank the reviewer for raising this point. Please see our common response on relaxation to [a, b].
>
> ---
>
> > 6. I think there is a change in notation when comparing Equations (1),(2),(3) and Figure 1 and the paragraph above it: While in the equations the kth order distribution is donoted by F^k, in the figure and its description it is denoted by F_k.
>
> We thank the reviewer for pointing this out. We have updated the notations in Figure 1 and its description to keep consistent with the rest of paper.
>
> ---
>
> > 7. I find the explanations surrounding equation (6) somewhat confusing: The equation itself specifies the condition that $X_{\theta^\star}$ is not SD-dominated by $X_{\theta}$. Below this, it is stated that $X_{\theta^\star}$ is not dominated by any other variable. It seems to me that an all-quantifier is missing before equation (6) ($\forall \theta$). Only then does the reformulation at the beginning of Section 3.1 make sense.
>
> We thank the reviewer for pointing this out. We have updated the revision to include the missing all-quantifier $\forall \theta$.
>
> ---
>
> > 8. The idea of the relaxation in Equation (7) seems to be close to the notion of almost SD, just replaced by almost non-SD. While I think the idea is nice in principle and can understand it from an optimization standpoint, I still wonder how strong the statement is in the end: Especially when combined with the truncated tails from equation (5), it seems to me that a lot of precision is lost (especially since Theorem 2 also only provides an almost undominated solution). A little discussion on this would certainly be enlightening.
>
> Please see our common response on the performance loss due to relaxation to [a, b].
>
> ---
>
> > 9. The reformulation of condition (6) at the beginning of Section 3.1 seems a little sloppy to me. I understand that a maximum element $X_{\theta^\star}$ can be characterized equivalently by requiring that $\min_{\theta \in \Theta} \Omega(X_\theta, X_{\theta^\star}) \ge 0$.
> However, no requirement is made for the $\arg\min$ specified here. Perhaps one could describe in more detail that this $\arg\min$ is a candidate for an undominated solution. If one starts with a dominated $X_{\theta^\star}$, one still finds an $\arg\min$, but this does not satisfy the condition.
>
> We appreciate the reviewer for the feedback. Incoporating $\theta \text{ is undominated}$ in $\arg\min \Omega_k(X_\theta, X_{\theta_t})$ requires an oracle that solves the original problem which sounds like an endless loop. We remark that if there are two solutions $Y_1$ and $Y_2$ where $Y_2$ dominates $Y_1$ by $\epsilon > 0$, we have $\Omega_k(Y_2, X_{\theta_t}) \le \Omega_k(Y_1, X_{\theta_t}) - \epsilon$ and hence $Y_2$ will be picked in favor of $Y_1$.
>
> ---
>
> > 10. I believe the symbol x_theta (lowercase x), which is first used in the bottom paragraph on page 7, was never formally defined.
>
> We thank the reviewer for the feedback. We defined $x_i$ following equation (14), at the first line on page 8:
> > ... (14), where $\\\{x_i\\\}\_{i=1}^N$ are N data points sampled from $X_\theta$. Here, we omit the dependency with $\theta$ in  $x_i$ for notational simplicity.

---

### Review · Reviewer_dCTY · 2025-10-23

**Summary Of Contributions:**

The authors provide an algorithm to find an approximate non-dominated solution satisfying relation (7)

**Audience:**

No

**Audience Explanation:**

Satisfying (7) is extremely weak. It simply means that $\theta^*$ is non-dominated.

Almost every $\theta^*$ that is optimal for an expected utility model will be non-dominated. By minimizing any coherent risk measure, we obtain non-dominated solutions. If we just maximize (or minimize in case of a loss) the expected value, we will obtain a non-dominated solution.

Unless the authors show that their method selects particular non-dominated solutions enjoying some desirable properties, this is a non-result. In the now-standard approach, we require that the solution dominate a benchmark and optimize (under this condition)
the expected performance. It is more rational. We obtain a non-dominated solution that compares favorably to a benchmark.

**Broader Impact Concerns:**

I do not see any relevant issues here.

**Claims And Evidence:**

Yes

**Claims Explanation:**

The authors correctly calculate subgradients of $\Omega$ and construct an algorithm

**Requested Changes:**

Prove that your method finds particular nondominated solutions, for example, better in some sense than a benchmark, or some other idea.

---

> ### Author Response · Authors · 2025-11-01
> **Response to Reviewer dCTY**
>
> We thanks the reviewer for the feedback. While optimizing a coherent risk measure along leads to non-dominated solutions in some cases, this is not generally true.
> - We have shown in the RL experiment that CVaR baselines are stochastically dominated by our proposed method in second order, except for the optimally tuned $\alpha$.
> - We can further show rigorously that optimizing $\mathrm{CVaR}\_{\alpha}$ can fail regardless of $\alpha$. Expanding on the toy example we mentioned in the common response on the comparison with CVaR baselines: Let $\theta\in[0,1]^2$ be a two-dimensional vector, with each of its element controlling the width of a uniform distribution. Specificallly, we design $X_\theta$ to be continuous random variable with it density function given by
> $$
> f_{X_\theta}(\eta) =
> \begin{cases}
> \frac{1}{2[\theta]_1}, & -\frac{1}{2}(1+[\theta]_1) \le \eta \le -\frac{1}{2}(1-[\theta]_1), \\\\
> \frac{1}{2[\theta]_2}, & \frac{1}{2}(1-[\theta]_2) \le \eta \le -\frac{1}{2}(1+[\theta]_2), \\\\
> 0 & \text{otherwise}.
> \end{cases}
> $$
> For $[\theta]_1 = 0$ and $[\theta]_2=0$ we simply set the corresponding density to a Dirac measure. We plot the density function and the $F^2$ curve of $X\_{\theta}$ for $\theta = [1,1],[0.5,0.8],[0.6,0.4],[0.01, 0.01]$ and upload the figure to the anonymous link
> https://ibb.co/WdwsRBt. With simple calculation one can show that
>   - $X_{\theta_1}$ stochastically dominates $X_{\theta_2}$ in second order if $\theta_1 \preceq \theta_2$;
>    - $$
> \mathrm{CVaR}\_{\alpha} (X_\theta) =
> \begin{cases}
> -\frac{1}{2} + (\alpha-\frac{1}{2})[\theta]_1, & 0 \le \alpha \le 0.5, \\\\
> \frac{1}{2} - \frac{1}{2\alpha} + (\alpha+\frac{1}{2\alpha}-\frac{3}{2})[\theta]_2, & 0.5 < \alpha \le 1. \\\\
> \end{cases}
> $$
>
> While $\text{CVaR}_{\alpha}$ is a coherent risk measure, a single choice of $\alpha$ only involves one of $[\theta]_1$ and $[\theta]_2$, and hence optimizing it does not lead to $\theta^\star = [0,0]$.

---

> ### Comment · Reviewer_dCTY · 2025-11-01
> **It is for the authors to prove that their result is not simply non-dominated**
>
> The authors are right that I was not precise enough by stating that the minimization of any coherent risk measure leads to non-dominated solutions. When the minimizer is not unique, some minimizers may be dominated by other minimizers. Minimizing a coherent risk measure that depends on the entire distribution, not just the worst quantiles, like CVaR, always leads to nondominated solutions. For example, a mix of CVaR with the expected value, mean-semideviation, entropic risk, etc.
>
> The main task for the authors is, though, not to find an irrelevant inaccuracy in the review, but to demonstrate the validity of their own result. Do they claim that the solution obtained by their approach dominates a minimizer of any CVaR? What about the simplest CVaR with alpha=1, which is the expected value? It is impossible to stochastically dominate the minimizer (or maximizer, depending on the setting) of the expected value. Perhaps they have a limit on alpha? I do not see such results in the manuscript.
> I have to reiterate the main issue: show that their solution is not only non-dominated, which is easy to achieve, but has some other useful properties.

---

> > ### Author Response · Authors · 2025-11-02
> >
> > It is our understanding that the reviewer believes our work is a non-result because finding a non-dominated solution is trivial and can easily achieved by using/combining existing approaches, as claimed initially by the reviewer any coherent risk measure leads to non-dominated solutions. We respectfully disagree and constructed the example to debunk this claim, and don’t view this as correcting "irrelevant inaccuracy" since this is quite central to the reviewer’s claim.
> >
> > The reviewer now revised this to “a coherent risk measure that depends on the entire distribution.” To our knowledge, there is no formal theorem or established method guaranteeing that minimizing such a risk measure necessarily yields a non-dominated solution. We would greatly appreciate it if the reviewer could point us to any references supporting this claim.
> >
> > The reviewer also believes that the standard "stochastic dominance constraint optimization" approach, or simply optimizing the expected value can lead to a non-dominated solution. The toy example in our previous response shows otherwise. In that example, all $\theta$ yield the same expected value (or $\mathrm{CVaR}_{1}=0$), yet only $\theta^\star$ stochastically dominates the others. Therefore, minimizing the expected value alone does not ensure non-dominance.
> >
> > The reviewer asks us to “demonstrate the validity of their own result” by answering “Do they claim that the solution obtained by their approach dominates a minimizer of any CVaR?“. However, this is something we have never claimed in our paper, and it is puzzling why we have to claim such a result to be valid. While the reviewer certainly can hold their own preference about which risk-sensitive solution is the best, we are also not claiming our risk-sensitive solution is the best (i.e., it dominates all other risk-sensitive solutions); indeed, the goal of our work is to promote the diversity of risk-sensitive solutions by offering computational tractable algorithms for stochastic dominance.
> >
> > We believe that our algorithm that finds a non-dominated solution is an important contribution, and the applications showcased in the paper also validate the usefulness of our algorithm. While we agree with the reviewer further results and research are greatly welcomed in this space, they are out of the scope of the current paper, whose results are substantial on its own. The philosophy of TMLR is to “to “emphasizes technical correctness over subjective significance”, and we kindly ask the reviewer to judge our paper on its own ground, not for claims we are not making.

---

> > ### Author Response · Authors · 2025-11-02
> >
> > Since the reviewer enters the party late, it is possible that you missed some of our earlier responses. In particular, the earlier Common Response [2/2] that addressed the non-uniqueness and comparisons to CVaR baselines might be of interest.

---

> ### Comment · Reviewer_dCTY · 2025-11-02
> **consistency of risk measures with stochastic dominance**
>
> I want to draw the attention of the Authors to the following argument. Suppose we are in a maximization setting and $\theta'$ maximizes $E[ X_{\theta}]$. If $X_{\theta'}$ were strictly dominated in the second order by some $X_\beta$, then $E[X_{\beta}] \ge  E[X_{\theta'}]$. Therefore, $\beta$ is equally good as $\theta'$. If the maximum is unique, a contradiction arises.
>
> For a more general case, suppose $\theta'$ minimizes a law-invariant coherent measure of risk $\rho(X_\theta)$, but $X_{\theta'}$ is strictly dominated in the second order by some $X_\beta$. Then it follows from the consistency of law-invariant coherent risk measures with stochastic dominance that we must have $\rho(X_\beta) \le \rho(X_{\theta'})$. This is a well-known result; see, e.g., Thm. 5.22 in the recent book by Dentcheva and Ruszcynski, "Risk-Averse Optimization and Control" (Springer, 2024), as well as related results.
>
> Furthermore, strict dominance implies that
> $CVaR_p(X_{\beta}) < CVaR_p(X_{\theta'})$
> for a nontrivial interval $A$ of probabilities $p$ (this immediately follows from the extremal representation of CVaR). So, if the risk measure $\rho()$ has a Kusuoka representation with measures $\lambda$ assigning positive probability to $A$, we get $\rho(X_{\beta}) < \rho(X_{\theta'})$, a contradiction.
>
> Summing up, if we minimize a law-invariant coherent measure of risk which depends on all CVaRs of the outcome, such as a spectral risk measure with full spectrum, or the mean-semideviation risk measure, we obtain a nondominated solution. For other law-invariant coherent risk measures, a nondominated solution is also a minimizer.
>
> The same can be said about maximizing $E[u(X_\theta)]$, with a nondecreasing and concave utility function $u()$. A nondominated solution is among the maximizers, and if $u()$ is strictly concave, the maximizer is nondominated.

---

> > ### Author Response · Authors · 2025-11-04
> >
> > We thank the reviewer for the careful and technically informative comment. The reviewer has made two main points:
> >
> > 1. Assuming a risk measure that is consistent with stochastic dominance has a unique optimal solution, then it is guaranteed to be non-dominated.
> > 2. In general, non-dominance can be achieved by optimizing law-invariant coherent measure of risk which depends on all CVaRs of the outcome, such as a spectral risk measure with full spectrum, or the mean-semideviation risk measure, or simply $E[u(X_\theta)]$ any nondecreasing and strictly concave utility function $u()$.
> >
> > Regarding 1, while we have never denied that finding non-dominated solutions can be easily achieved in some scenarios, assuming uniqueneess sounds like an over-simplification and should not undermine this work as a general and practical method achieving non-dominance.
> >
> > Regarding 2, we would like to first thank the reviewer for pointing out that optimizing $E[u(X_\theta)]$ with nondecreasing and strictly concave $u$ guarantees non-dominated solution. We will revise the manuscript to state this precisely. However, we remark that
> > 1. The utility function needs to be properly selected to accomodate the distribution of $X_\theta$, e.g., the gradient of $u()$ needs to be sensitive in the regions of the outcome distribution. Our algorithm, by contrast, adapts its update to the observed distribution and therefore is less sensitive to a priori tuning of a single utility or spectrum.
> > 2. While optimizing $E[u(X_\theta)]$ as a single objective is fine, it cannot in general be applied jointly with other risk measures without distorting the preference. On the other hand, our optimization formulation (8) leads to zero gradient when the algorithm reachs any non-dominated solution. This unique property allows our method to be combined with arbirary choice of risk measure optimization and guarantees non-dominance without distorting the original preference, as long as the risk measure itself is consistent with stochastic dominance. (See our common response on non-uniqueness and interpretation of the outcome).
> >
> >
> > The reviewer also suggests approaches like optimizing $\int_0^1 \mathrm{CVaR}_\alpha\mathrm{d}\alpha$. We have mentioned this in Related Works section but it is not clear how this leads to a practical and efficient algorithm design. In contrast, we provide complexity analysis to our algorithm to guarantee its efficiency.
> > > Remarkably, second-order stochastic dominance (SSD) can be interpreted as a continuum of $\mathrm{CVaR}_\alpha$ comparison over the entire risk level set (0, 1) (Martin et al., 2020)
> >
> > We want to wrap up the response by drawing an analogy to this debate as food for thought: are there values in developing new algorithms for nonconvex optimization with only first-order convergence guarantees? Indeed, it is often the case we may only (initially) provide first-order convergence guarantees of new algorithms to stationary points due to the highly nonconvex nature of the problem. However, it does not mean such new algorithms are not useful because "finding first-order stationary points are trivially done by gradient descent". We believe our framework develops an alternative risk-sensitive criterion by stochastic dominance, which has never been exploited, and our algorithm and a thorough treatment to its various case studies in applications provide a meaningful and novel contribution to the important problem of risk-sensitive learning and optimization. While it will be interesting to further study the solution structures in future works, such additional quest should not diminish the value of the algorithm development in the current paper.

---

### Author Response · Authors · 2025-10-06
**Common Response [1/2]**

## **Common response on relaxation to [a, b]**

### **Why is it necessary to introduce [a, b]?**
The introduction of $[a, b]$ in this work is to address the technical difficulty of uniformly bounding the difference between the k-th distribution function $F^k_X$ and its empirical estimate $\widehat{F}^k_X$, in order to properly assess the solution's optimality in stochastic dominance. For $k = 1$, $F^k_X$ is simply the standard CDF and Dvoretzky–Kiefer–Wolfowitz inequality gives a uniform bound over $\eta \in \mathbb{R}$; For $k\ge 2$ we need either a restriction on the range of $\eta$ or a tail/moment condition on $X$ to derive a finite-sample bound.

We remark that restricting $\eta$ to $[a,b]$ (or simply assuming all $X$ are bounded within $[a,b]$/discrete) is standard practice in stochastic dominance literature, see e.g. [1-7].

[1] Ingersoll, Jonathan E. Theory of financial decision making. Vol. 3. Bloomsbury Publishing PLC, 1987.

[2] Dentcheva Darinka and Andrzej Ruszczynski. "Optimization with stochastic dominance constraints." SIAM Journal on Optimization, 14(2):548–566, 2003.

[3] Dentcheva Darinka and Andrzej Ruszczyński. "Optimality and duality theory for stochastic optimization problems with nonlinear dominance constraints." Mathematical Programming 99, no. 2 (2004): 329-350.

[4] Dentcheva Darinka and Andrzej Ruszczyński. "Portfolio optimization with stochastic dominance constraints." Journal of Banking & Finance 30, no. 2 (2006): 433-451.

[5] Eeckhoudt, Louis, Harris Schlesinger, and Ilia Tsetlin. "Apportioning of risks via stochastic dominance." Journal of Economic Theory 144, no. 3 (2009): 994-1003.

[6] Chan, Raymond H., Ephraim Clark, Xu Guo, and Wing-Keung Wong. "New development on the third-order stochastic dominance for risk-averse and risk-seeking investors with application in risk management." Risk Management 22, no. 2 (2020): 108-132.

[7] Luo, Chunling, Piao Chen, and Patrick Jaillet. "Portfolio optimization based on almost second-degree stochastic dominance." Management Science 71, no. 8 (2025): 7029-7055.

### **How much performance loss due to $\eta \in [a, b]$?**

Assume a solution $X_{\theta^\star}$ is not dominated in $[a, b]$ and also bounded within $[a, b]$, then it is not dominated over the entire real line $\mathbb{R}$, i.e., no "performance loss". To see this, note that any $Y$ that dominates $X_{\theta^\star}$ over $\mathbb{R}$ must dominate $X_{\theta^\star}$ over $[a, b]$ as well. Therefore, we have $F_{Y}^k(\eta) = F_{X_{\theta^\star}}^k(\eta)$ and $F_{Y}(\eta) = F_{X_{\theta^\star}}(\eta)$ for $\eta\in[a,b]$. As $F_{X_\theta^\star}(a) = 0$ and $F_{X_{\theta^\star}}(b) = 1$, we conclude that $Y$ must be equal to $X_{\theta^\star}$ in distribution.


If $X_{\theta^\star}$ has support larger than $[a, b]$, then the "performance loss" depends on the properties of tails of the distribution outside of [a, b].

### **What would happen if [a, b] is replaced with $\mathbb{R}$ in algorithm implementation?**

This amounts to solve $\arg\max_{\eta\in\mathbb{R}}\widehat{F}^k_X(\eta) - \widehat{F}^k_Y(\eta)$. For $k=2$ (see Algorithm 2), the objective $\widehat{F}^k_X(\eta) - \widehat{F}^k_Y(\eta)$ is linear on interval $[\eta_i, \eta_{i+1}]$ for $i=1,...,2N-1$, is $0$ on interval $(-\infty, \eta_1]$ and has slope=0 on interval $[\eta_{2N}, \infty)$. Therefore the maxima is achieved in $\{\eta_i, i=1,...,2N\}$. For $k\ge 3$ some extra effort needs to be made to take care of the polynomial maxima on $[\eta_{2N}, \infty)$. To summarize, this results in minimal change to the algorithm but the performance guarantee would now depend on the tail/moment condition of $X$'s distributions.

---

### Author Response · Authors · 2025-10-06
**Common Response [2/2]**

## **Common response on non-uniqueness and interpretation of the outcome**

By design the uniqueness is only guaranteed when there is a optimal solution $\theta^\star$ dominating all other $\theta\in\Theta$. To add more interpretation/controllability, a straightforward approach is to combine the algorithm's update with standard update. For example, due to the form of $g_{t,\bar{t}}$ as dynamically weighted SGD in algorithm 1, one can combine the update with ERM's update by modifying sample $x_{t,\bar{t},i}$'s weight from $\partial_{x_{t,\bar{t},i}}\widehat{u}^\star(x_{t,\bar{t},i})$ to
$$\frac{\lambda\partial_{x_{t,\bar{t},i}}\widehat{u}^\star(x_{t,\bar{t},i})+1}{\lambda+1}$$
for some $\lambda \ge 0$. We recover ERM's SGD step with $\lambda = 0$ and LSD's update with $\lambda=+\infty$. The combined update is "compatible" in the sense that if $X$ stochastically dominates $Y$ then we have $\mathbb{E}[X] \ge \mathbb{E}[Y]$, so that ERM's update will not contradict LSD's update. This combined update would guide the model towards a solution that both minimizes empirical risk and is not dominated.

---
## **Common response on the comparison with CVaR baselines**

We have added the numerical results of CVaR-PG [1] in our modified CartPole experiment. We remark that some CVaR RL algorithms (e.g. [2], chapter 7.8 [3]) are based on value-iteration and are not directly applicable to RL with continuous states. The results show that CVaR-PG can achieve performance comparable to LSD, but only with a carefully tuned choice of $\alpha$, since $\mathrm{CVaR}_{\alpha}$ ignores distributional properties above the chosen quantile.

Moreover, one can construct examples—such as rewards drawn from a mixture of non-overlapping truncated Gaussian distributions with variances controlled by the policy—where optimizing $\mathrm{CVaR}_\alpha$ only affects the component associated with the lower $\alpha$-quantile, leaving the others unoptimized. In contrast, stochastic dominance criteria account for the entire distribution, offering a more holistic measure of risk sensitivity.

[1] Tamar, Aviv, Yonatan Glassner, and Shie Mannor. "Optimizing the CVaR via sampling." In Proceedings of the AAAI Conference on Artificial Intelligence, vol. 29, no. 1. 2015.

[2] Chow, Yinlam, et al. "Risk-sensitive and robust decision-making: a cvar optimization approach." Advances in neural information processing systems 28 (2015).

[3] Bellemare, Marc G., Will Dabney, and Mark Rowland. Distributional reinforcement learning. MIT Press, 2023.

---

### Decision · Action_Editor_ZkV1 · 2025-11-13

**Recommendation:** Accept with minor revision

**Additional Comments:**

Reviewer `dCTY` strongly requests making sure that the final version explains that other methods for finding non-dominated solutions exist.

**Audience:**

Yes

**Audience Explanation:**

We had an unusual review process, and some hearty discussions. The chief sticking point in discussions (raised by `dCTY`) was that just because the algorithm finds a non-dominated solution, is that necessarily helpful, since there are many non-dominated solutions and some are useless. But after the discussions and rebuttals, all authors seem to agree that there is some benefit to the result. Quoting `dCTY`, "The paper contributes to our knowledge, and the discussion was useful." In particular, the paper not only introduces and analyzes an algorithm, but also introduces a new criterion based on stochastic dominance in a manner that has not been done before.

Quoting reviewer `6wf5`, "In addition, I believe that the further spread of SD-based reasoning in machine learning is a promising field, for which this article raises awareness."

**Claims And Evidence:**

Yes

**Claims Explanation:**

The paper has both math and a rigorous analysis, as well as numerical results. No reviewers raised issues with the technical proofs. There was some discussion of empirical evidence, and the revision added more results. In the end, the reviewer and AE are satisifed about the correctness.

---

> ### Author Response · Authors · 2025-11-20
>
> Thank you so much for your exemplary shepherding of the review process! We appreciate the useful feedbacks and will incorporate them into the final paper.